# Stem cell culture conditions affect *in vitro* differentiation potential and mouse gastruloid formation

Marloes Blotenburg[1,2]*, Lianne Suurenbroek[1,2], Danique Bax[3], Joëlle de Visser[4], Vivek Bhardwaj[1,2], Luca Braccioli[5], Elzo de Wit[5], Antonius van Boxtel[4], Hendrik Marks[3], Peter Zeller[1,2,6]*

**1** Hubrecht Institute-KNAW (Royal Netherlands Academy of Arts and Sciences), Oncode Institute, Utrecht, The Netherlands, **2** University Medical Center Utrecht, Utrecht, The Netherlands, **3** Department of Molecular Biology, Faculty of Science, Radboud Institute for Molecular Life Sciences (RIMLS), Radboud University Nijmegen, Nijmegen, The Netherlands, **4** Developmental, Stem Cell and Cancer Biology, Swammerdam Institute for Life Sciences, University of Amsterdam, Amsterdam, The Netherlands, **5** The Netherlands Cancer Institute, Amsterdam, The Netherlands, **6** Department of Molecular Biology and Genetics, Aarhus University, Aarhus, Denmark

☯ These authors contributed equally to this work.
* peter.zeller@mbg.au.dk (PZ), marloes.blotenburg@unil.ch (MB)

## Abstract

Aggregating low numbers of mouse embryonic stem cells (mESCs) and inducing Wnt signalling generates 'gastruloids', self-organising complex structures that display an anteroposterior organisation of cell types derived from all three germ layers. Current gastruloid protocols display considerable heterogeneity between experiments in terms of morphology, elongation efficiency, and cell type composition. We therefore investigated whether altering the mESC pluripotency state would provide more consistent results. By growing three mESC lines from two different genetic backgrounds in different intervals of ESLIF and 2i medium the pluripotency state of cells was modulated, and mESC culture as well as the resulting gastruloids were analysed. Microscopic analysis showed a pre-culture-specific effect on gastruloid formation, in terms of aspect ratio and reproducibility. RNA-seq analysis of the mESC start population confirmed that short-term pulses of 2i and ESLIF modulate the pluripotency state, and result in different cellular states. Since multiple epigenetic regulators were detected among the top differentially expressed genes, we further analysed genome-wide DNA methylation and H3K27me3 distributions. We observed epigenetic differences between conditions, most dominantly in the promoter regions of developmental regulators. Lastly, when we investigated the cell type composition of gastruloids grown from these different pre-cultures, we observed that mESCs subjected to 2i-ESLIF preceding aggregation generated gastruloids more consistently, including more complex mesodermal contributions as compared to the ESLIF-only control. These results indicate that optimisation of the mESCs pluripotency state allows the modulation of cell differentiation during gastruloid formation.

**Data availability statement:** All data generated for this project, code used and annotated notebooks for performing the analysis and generating figures can be found on github (https://github.com/marloes3105/preculture). The raw data was deposited at the GEO data repository. The accession codes are GSE268783 for the single-cell RNA sequencing data of mESCs and gastruloids, GSE268814 for the bulk DNA methylation data of mESCs, and GSE268816 for the bulk profiling of H3K27me3 in mESCs.

**Funding:** P.Z. was funded by SNF (P2BSP3-174991), HFSP (LT000209/2018-L) and Marie Skłodowska-Curie Actions (798573). V.B. was funded by EMBL LTF (ALTF 1197–2019). D.B, A.v.B., J.d.V., E.d.W. and H.M. were funded by an ENW XL (OCENW.XL21.XL21.100). L.B. and E.d.W were supported by ERC Consolidator grant (865459, 'FuncDis3D') and Vidi grant '016.16.316' from the Dutch Research Council (NWO). This work is part of the Oncode Institute which is partly financed by the KWF Dutch Cancer Society. There was no additional external funding received for this study.

**Competing interests:** The authors have declared that no competing interests exist.

# Introduction

The first mouse embryonic stem cell (mESC) line, derived from the inner cell mass of the pre-implantation blastocyst, was established in 1981 [1,2]. Over the past decades, these pluripotent cells have become a powerful tool for studying and understanding cellular differentiation during embryonic development. This is illustrated by the fact that an ever growing number of specific cell types can now be generated through directed differentiation [3–5]. In addition to the commonly used two-dimensional differentiation, it is well established that aggregation of mESCs into three-dimensional (3D) spheres termed embryoid bodies (EBs), causes their spontaneous differentiation into structures containing cell types derived from all three embryonic germ layers [6,7]. Although EBs are mostly unorganised, a small fraction spontaneously breaks symmetry in a stereotypical manner and acquires a spatial organisation that resembles gastrulating embryos [8]. This remarkable capacity to self-organise can be more consistently achieved by aggregating 300-600 mESCs followed by incubation with a Wnt-activator (Chiron) from 48 - 72 hrs [9–11]. At 120 hrs, the resulting structures, termed 'gastruloids' indeed contain cell types derived from all three germ layers, organised into three body axes and contain collinear Hox gene expression [12–16]. Their amenability to genetic and chemical perturbations, single cell sequencing approaches and imaging has made gastruloids an increasingly popular tool for studying poorly understood aspects of early mammalian embryogenesis *in vitro*.

Although gastruloids have already yielded important insights, there is a need to improve the current models. Next to the lack of extra-embryonic tissues, gastruloids are largely devoid of organised anterior structures such as anterior mesoderm and brain precursors [12,14,15]. However, through the addition of signalling molecules, gastruloids can be directed to form these cell types. For instance, anterior structures can be induced through the addition of FGF and Activin A and replacement of Wnt-activation with Wnt-inhibition. Similarly, early stages of heart development can be modelled by including cardiogenic factors FGF, ascorbic acid, and VEGF [17,18]. Lastly, through embedding gastruloids in matrigel before the elongation stage, tissue structures such as somites, the neural tube, and the gut tube are more faithfully reproduced [15,19–21]. Together these studies demonstrate that by optimising culture conditions, a wider variety of cell types and tissue structures can be induced in gastruloids.

In addition to a lack of anterior and extra-embryonic cell types, gastruloid formation using current protocols can be highly variable within experiments. This results in unpredictable frequencies of cells belonging to specific lineages and also in variable spatial organisation [22]. Such inter-gastruloid heterogeneity poses obvious challenges with respect to interpretation of results and reproducibility and therefore warrants further exploration and optimisation. Inter-gastruloid variability seems, at least in part, caused by culture conditions used prior to gastruloid formation, which we term "pre-culture". Generally, mESCs are maintained in a medium containing serum, termed ESLIF medium, which gives rise to a heterogeneous pool of cells in terms of their transcriptional state [23]. ESLIF-grown cells are most comparable to peri-implantation epiblast cells and are commonly referred to as being in a naive pluripotency state [24]. Alternatively, cells can be maintained in serum-free medium containing Glycogen Synthase Kinase 3-beta (GSK3b) and MAPK/ERK kinase (MEK) inhibitors, commonly referred to as 2i medium. Cells grown in 2i are more homogeneous and correspond to stem cells found in the inner cell mass of the pre-implantation embryo, commonly referred to as ground-state pluripotency [23,25]. The main differences between ESLIF and 2i-grown mESCs are related to their epigenome. In particular, DNA methylation and the histone modification H3K27me3, indicative of a repressive chromatin state, show variable distributions between the two conditions. 2i-grown mESCs show a genome-wide DNA methylation level of approximately 30% and a general spread of H3K27me3 across the genome [25–27]. In contrast, in

serum-grown mESCs, DNA methylation covers 80% of the genome, and H3K27me3 shows more focused distributions around promoter regions [25,28–31]. The epigenome and therefore the pluripotency states are dynamic and ES cells can transition from one state to another through intermediate states [32]. Thus, optimising pre-culture conditions and linking these conditions to their epigenetic and transcriptional states will likely result in a more robust gastruloid model. Here, we explored pre-culturing mESC using different combinations of 2i and ESLIF culture media, profiled their pluripotency state, and studied the effects on gastruloid differentiation and lineage contributions. Our data highlight the importance of carefully considering pre-culture conditions in relation to the mESC line used.

## Materials and methods

### Cell culture and gastruloid generation

129S1/SvImJ/ C57BL/6 (B6; [33]), 129/Ola E14-IB10 (IB10; [34]) and E14-triple reporter (TR; [35]) mESCs were cultured in a humidified incubator (5% $CO_2$, 37 °C) on 0.1% (B6) or 0.15% (IB10, TR) gelatin-coated 6-well plate cell culture dishes in ESLIF medium (GMEM (B6; Gibco) or DMEM (IB10, TR; Gibco) containing 10% (B6) or 15% (IB10, TR) fetal bovine serum (FBS, Sigma F7524 (B6), Bodinco #5010 (IB10, TR)), 1 mM Sodium Pyruvate, 1% non- essential amino acids (Gibco), 1% GlutaMAX supplement (Gibco), 1% penicillin-streptomycin (Gibco), 0.1 mM b-mercaptoethanol, and 1000 units/mL mouse leukaemia inhibitory factor (mLIF, ESGRO (B6) or home made (IB10, TR)). Cells were split every second day at 80% density by washing with PBS0 (PBS without calcium or magnesium), dissociating with TrypLE (B6; Gibco, 12605010) or 0.05% trypsin-EDTA (IB10, TR, B6 stainings in S1J Fig; Gibco, 25300096) for ~ 5 mins at 37 °C, centrifuging at 300 g for 3 mins, and replating the pellet 1:5 (B6) or 1:10 (IB10, TR) on gelatinised plates. Pre-culture conditions consisted of different pulse timings and lengths with 2i medium (For B6 cells: 48.1% DMEM/F12 (Gibco) and 48.1% Neurobasal (Gibco) containing 0.5% N-2 supplement (Gibco, # 17502048), 1% B-27 supplement (Gibco, # 17504044), 1% GlutaMAX, 1.1% penicillin-streptomycin, 0.1 mM b-mercaptoethanol, 1000 units/mL mouse leukaemia inhibitory factor (mLIF, ESGRO), 3 μM chiron (CHIR99021, Tocris # 4423) and 1 μM PD032509 (Sigma, # PZ0162), for IB10 and TR cells: NDiff 227 (Takara, Y40002), 1% penicillin-streptomycin (Gibco), 3 μM Chiron (CHIR99021, Selleckchem S1263), 1 μM PD032509(Selleckchem, S1036) and mLIF (home made)). During pre-culture conditions, cells were split at day 1 and 3; medium was refreshed at day 2 and 4. All cell lines were routinely tested and confirmed to be free of mycoplasma.

Gastruloids were generated as described before [12,15] with the following adaptations for B6 cells: two days before aggregation, cells were plated in a series of 1:10 to 1:3 dilution, and at the time-point of aggregation the cells with 80% confluency were chosen. Further, N2B27 medium (Takara) was passed through a.22 μm filter before use and 600 cells were used for aggregation instead of 300. For gastruloids developed for immunofluorescence stainings (S1J Fig), N2B27 medium was made as follows: 48.1% DMEM/F12 (Gibco) and 48.1% Neurobasal (Gibco) containing 0.5% N-2 supplement (Gibco, # 17502048), 1% B-27 supplement (Gibco, # 17504044), 1% GlutaMAX, 1.1% penicillin-streptomycin, 0.1 mM b-mercaptoethanol. For IB10 and TR cells, gastruloids were generated with the following adaptations: live cells were counted by using 0.2% Trypan Blue solution (Gibco) and a LUNA-FX7 cell counter (Logos biosystems) and 300 live cells were used for aggregation.

### Bright-field microscopy and analysis

B6 gastruloids were kept in 96-well plates and imaged using the Leica THUNDER Imager Live Cell & 3D Cell Culture & 3D Assay with a 10x dry lens. During imaging, the temperature was

kept at 37 °C and the CO2 level at 5%. IB10 and TR gastruloids were imaged in 96-well plates using a Zeiss Axio Vert.A1 with a 5x objective. For B6 gastruloids, multiple Z-stacks were taken of each gastruloid with an interval of 5 μm and Z-stacks were focused using the ImageJ data processing package Fiji (v.2.1.0/1.53c) plug-in Stack Focuser. For all cell lines, images were subsequently analysed by creating a mask of the gastruloid in Fiji (v2.14.0). The pictures and masks were loaded into the MOrgAna software [36] for gastruloid quantifications of the major and minor axis length. Each gastruloid was straightened and the aspect ratio was determined by dividing the major and minor axis lengths. Statistics were calculated with statannot's add_stat_annot (v0.2.3). For comparison of 2 conditions an independent T-test was used, and for more than two conditions the Mann-Whitney-Wilcoxon two-sided test with Bonferroni multiple testing correction was used.

## Immunofluorescence and confocal microscopy

Gastruloids were harvested, fixed in 4% PFA and stored in PBST (PBS, 0.1% Tween20) until further processing. For immunofluorescent staining, gastruloids were washed three times for 5 minutes in PBST, followed by blocking for 1h in blocking buffer (BB; PBS, 10% fetal bovine serum (FBS), 0.2% Triton). Gastruloids were incubated overnight at 4°C with Rabbit anti-TBXT (1:300, Abcam, #ab209665) or Goat anti-TBXT (1:300, R&D systems, #AF2085) and Rat anti-SOX2 (1:300, ThermoFisher, #14-9811-82) primary antibodies in BB. Antibody specificity was determined by omitting primary antibodies. Stained gastruloids were washed three times for 10 minutes in BB followed by incubation with Donkey anti-Rabbit AF647 (1:500, Invitrogen, A31573) or Donkey anti-Goat AF568 (1:500, Invitrogen, #A11057) and Donkey anti-Rat AF488 (1:500, Invitrogen, #A21208) secondary antibodies in BB for 2 hours at RT. After a single wash in PBST for 5 minutes, gastruloids were counterstained with DAPI (1:5000, Invitrogen, D1306) in PBST for 15 minutes at RT, and mounted onto Superfrost microscope slides (Epredia, # 16261541) using Mowiol mounting medium.

Gastruloids were imaged using the Leica SP8 confocal microscope equipped with a Fluotar VISIR 25x NA0.95 water long distance objective. Multiple z-stack images were acquired at 1024x1024px or 512x512px resolution with a step size of either 4 or 7 μm. LAS X and Fiji were used for image visualization and processing.

## FACS analysis of pooled TR gastruloids

At 120 h AA, TR gastruloids were pooled and washed with PBS0, followed by dissociation with 0.5% trypsin-EDTA for ~ 5 mins at 37 °C. Single cells were resuspended with a P200 pipette, spun down at 200 g for 5 mins and resuspended in 500 μL PBS0. 7-AAD (Invitrogen, 00-6993-50) was added in a 1:1000 dilution to stain dead cells. Cell fluorescence of the three reporters was measured on an 8-color MACSQuant (Miltenyi Biotec) and analysed with FlowJo software (v10).

## Dissociation, FACS sorting and scRNA-seq of B6 mESCs and single gastruloids

mESCs were dissociated as described, re-suspended in PBS0 (PBS without calcium or magnesium) supplemented with 5% FBS, and filtered through a 35 μm filter (Falcon, 352235). Single gastruloids were washed with PBS0, followed by incubation with 50 μL trypsin without phenol red (Gibco) for ~ 5 mins at 37 °C. Gastruloids were mechanically broken up into a single-cell suspension by pipetting with a P200 pipette, diluted to 1 mL with PBS0 supplemented with 5% FBS, and filtered through a 35 μm filter (Falcon, 352235). Prior to fluorescence-activated cell sorting (FACS) using an Influx machine, DAPI was added to the cell suspension to assess

cell viability. Dissociated single cells were sorted into two 384-well plates for each pre-culture mESC condition, resulting in a total of 768 cells per condition. For each individual gastruloid, one 384-well plate was sorted and processed, and 5 gastruloids were sorted per condition. Libraries were prepared following the CEL-seq2 protocol [37]. The libraries were sequenced with the Illumina NextSeq 500 with a 2 x 75 bp or 1 x 75 bp kit (26 bp at read 1 and 61 bp at read 2).

## sortChIC and bisulfite sequencing of mESCs

H3K27me3 and DNA methylation was assayed in bulk from the same cell suspension as used for CEL-seq2. For bisulfite sequencing, cells were pelleted after dissociation by centrifuging at 300 g for 3 mins, followed by DNA extraction using the Qiagen DNeasy Blood & Tissue Kit (69504) according to the manufacturer's guidelines. DNA yield was quantified with Qubit dsDNA HS Assay Kit (10616763) and 100 ng DNA was used as input for the Zymo Research Pico Methyl-Seq Library Prep Kit (D5456). Abundance and quality of the final library were assessed by Qubit dsDNA HS Assay Kit (10616763) and Agilent Bioanalyzer HS DNA kit (5067-4626) and sequenced paired-end 2 x 150 bp using an Illumina NextSeq 500. For sort-ChIC, dissociated cells were ethanol-fixed and incubated with 1:100 anti-H3K27me3 antibody (NEB, 9733S) as described previously [38]. 1000 cells were sorted in a tube and processed as described, with all single-cell volumes multiplied by 100 [38]. Samples were sequenced paired-end 2 x 100 bp on an Illumina NovaSeq 6000.

## Transcriptome analysis

Raw fastq files were processed into count tables using the transcriptome Snakemake [39] pipeline of the SingleCellMultiOmics package (see https://github.com/BuysDB/SingleCell-MultiOmics/wiki/Transcriptome-data-processing, version 0.1.30). The pipeline performs demultiplexing for CEL-Seq2 barcodes with a hamming distance of 0 using SingleCellMul-tiOmics' demux.py and trimming with cutadapt [40] (version 4.1), followed by mapping with STAR [41] (version 2.5.3a) to the 129/B6 SNP-masked GRCm38 mouse genome (Ensembl 97). Feature counting was performed with featureCounts [42] (version 1.6.2) and reads were tagged and deduplicated, then transformed to a.csv file. Downstream analysis was performed with Scanpy [43] (version 1.8.1). Cells with less than 500 reads or less than 100 genes detected were filtered out. Genes detected in less than 2 cells were also removed. Cells with more than 40% or more than 20% mitochondrial reads were excluded for mESCs and gastruloid cells, respectively. Mitochondrial reads, Malat1, and ribosomal reads were excluded from further analysis. Counts were normalised to 500 transcripts per cell and logarithmized. The number of counts, percentage of mitochondrial reads, and cell cycle phase were regressed out and each gene was scaled to unit variance with a maximum of 10. Principal component analysis was performed with 10 neighbours and either the 30 or 40 highest principal components were used to generate the UMAP for mESCs or gastruloid cells, respectively. The data was clustered using the Leiden algorithm (scanpy.tl.leiden, resolution set to 0.5). Differentially expressed genes between clusters were determined using Scanpy's Wilcoxon rank-sum test. Differentially expressed genes between ESLIF and 2i pre-culture states were determined by grouping pre-culture conditions into pseudobulks with decoupler (version 1.6.0) followed by pydeseq2 (version 0.4.4)

To match the preculture scRNAseq dataset with a reference gastruloid time-course dataset from 72h AA until 168h AA [44], scanpy's PCA-based integration called ingest was used and the lineage, pseudotime and sampling time labels were integrated (sc.tl.ingest(adata, adata_ref, obs = ["lineage","pseudotime","hour"])).

### H3K27me3 and DNA methylation data analysis

WGBS-seq data was mapped using the snakePipes [45] WGBS mapping workflow (v 1.3.1) to GRCm38 (mm10) genome with default parameters. WGBS workflow applies paired-end trimming using fastp [46] (v0.20) with options -q 5 -l 30 -M 5, mapping using bwa mem [47] with options -T 40 -B 2 -L 10 -CM -U 100 -p and bwa-meth [48] (v0.2.2) for methylation encoding, followed by CpG methylation calling using methyldackyl (https://github.com/dpryan79/MethylDackel, v0.5.0) with options -mergeContext -maxVariantFrac 0.25 -minDepth 4. ChIC-seq data was mapped using the ChIC Snakemake pipeline [39] of the SingleCellMultiOmics package (version 0.1.30). The CPM-normalised coverage was computed in 100 bp bins using deepTools [49] (v3.0) bamCoverage with options -normalizeUsing CPM -skipNAs -ignore X MT. We used the mm10 gencode [50] (v23) annotation to define promoter (TSS + - 10 kb), gene body (signal > 10 kb downstream of TSS, only on genes longer than 10 kb), intergenic regions and repbase database (2022) to define repeat regions. WGBS signal (CpG methylation ratio) and ChIC signal (counts) were then aggregated on these grouped features. In order to compare ChIC signal between features, we additionally normalised the ChIC signal to the total length of the feature groups using FPM (features per million) defined as $FPM = A * (1/(sum(A))) * 10 \times 10^6$, where A = feature counts/kb. Finally, to classify genes based on the ChIC signal at the promoters, we aggregated counts on promoter regions using computeMatrix with options -bs 200 -missingDataAsZero -skipZeros, followed by plotHeatmap with options -kmeans 4 to get the 4 gene clusters. The normalised counts per cluster was then calculated as $CPM = A/(sum(A) * 10 \times 10^6)$, where A = feature counts.

## Results

### Different intervals of 2i exposure during mESC culture affect the reproducibility of gastruloid formation

To test the impact of preculture-defined mESC states on gastruloid composition, we applied different combinations of 2i and ESLIF culture media to mESCs and studied the effects on gastruloid formation (Fig 1A). First, we tested three mESC lines from two different genetic backgrounds: E14 cells containing a triple germ layer reporter (TR), a wild-type E14-IB10 line (IB10), and wild-type 129SV/Bl6 (B6) cells. We included two control conditions: one with ESLIF only (1) and one with 2i only (2), representing naive and ground state pluripotency, respectively, and show the expected morphologies, with flat and connected colonies in ESLIF medium and dome-shaped disconnected colonies in 2i medium (Figs 1A, S1A). Because it was previously reported that some cell lines benefit from a pulse of 2i for gastruloid formation (reviewed in [51]), we hypothesised that short-term pulses of 2i could modulate the mESCs differentiation potential, thereby affecting gastruloid formation. To this end, we tested 2 intermediate states by culturing the mESCs in ESLIF with a pulse of 2i for the last 48 (3) or 24 (4) hours before aggregation. The morphologies of mESCs subjected to the different pulses are in between the ESLIF and 2i state, with condition 3 forming ESLIF-type colonies with round edges and condition 4 resembling the ESLIF control (Fig 1A).

For all three cell lines and pre-culture conditions, mESCs were aggregated into gastruloids following the standard protocol which uses a Wnt-activation pulse between 48 h and 72 h after aggregation (AA) through the addition of chiron [12,15]. Gastruloids display morphological variation at 120h AA depending on the mESC pre-culture condition used (S1B Fig). To characterise these morphological differences for all cell lines and conditions in a high number of gastruloids in parallel, we performed a gastruloid elongation screening, which is indicative for efficient symmetry breaking and early lineage commitment of cells towards the three germ lines. In short, we imaged all gastruloids generated, extracted the major and minor axis length

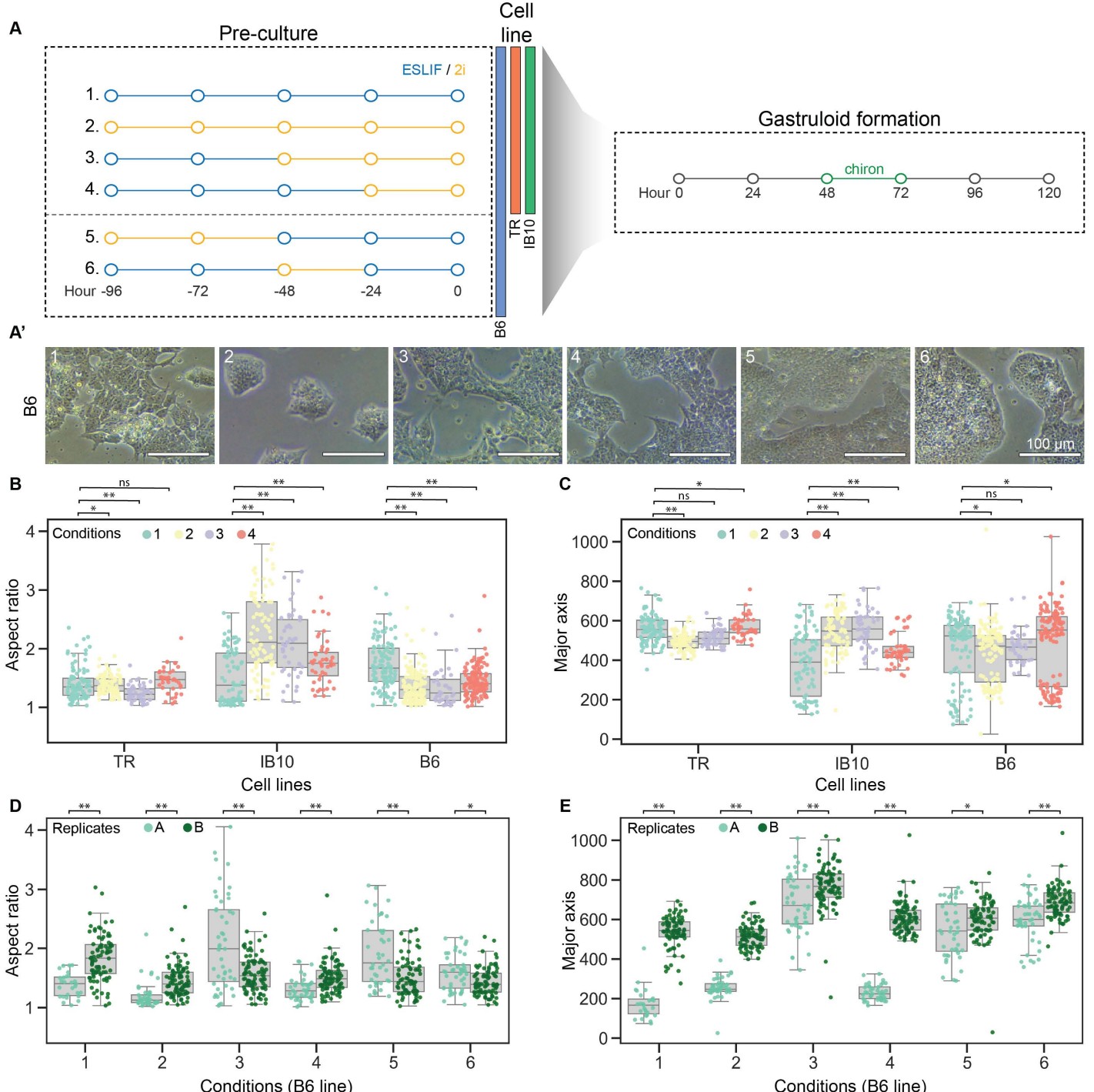

**Fig 1. Variations in mESC culture conditions affect gastruloid elongation length, efficiency, and consistency across replicates.** (A) Schematic overview of six different pre-culture conditions, with varying times of ESLIF or 2i culture medium, and (A') corresponding bright-field images showing cell morphologies for all conditions. All cells were routinely maintained in ESLIF medium on gelatin, with the exception of the 2i control (condition 2) in the TR and IB10 cell lines, which underwent full 2i reprogramming for 2 weeks. (B-C) Aspect ratio (B) and major axis length (C) of gastruloids from three different genetic backgrounds. (D-E) Aspect ratio (D) and major axis length (E) of gastruloids from B6 cell line, separated by replicate. The most efficient timing of chiron pulse and read-out was chosen for each condition (see S1F–G Fig). For condition 3, this was 24 h delayed (chiron pulse at 72-96 h AA, read-out at 144 h AA), for all other conditions, this was the standard protocol (chiron pulse at 48-72 h AA, read-out at 120 h AA). Statistical significance between replicates A and B was calculated with an independent t-test.

of each individual gastruloid using MOrgAna [36], and used these to calculate the aspect ratio (S1C Fig). Comparing results obtained for the 4 conditions showed a significant difference in gastruloid formation efficiency as measured by the aspect ratio (AR) and major axis length (Fig 1B–C). The gastruloids generated from the TR line displayed modest elongation and did not show significant differences between pre-culture conditions. In contrast, the IB10 and B6 gastruloids were affected by pre-culture conditions but with opposing effects. For IB10 gastruloids, 2i culture conditions resulted in a significant increase in AR whereas for B6 gastruloids, ESLIF culture conditions gave rise to the highest AR. These data demonstrate the variability of gastruloid formation between cell lines, and the requirement of different pre-culture conditions for gastruloid formation dependent on the cell line.

To see if pre-culture conditions affect lineage distribution in gastruloids, we further analysed formed gastruloids from the TR cell line for pre-culture conditions 1 and 2 by FACS measurement of the fraction of cells containing germ layer-specific expressed fluorescent proteins. We noticed variations in the ratio of cells expressing each marker (S1D–E Fig). While both conditions had similar fractions of endodermal cells, ESLIF pre-cultured gastruloids contained more ectodermal cells, while 2i pre-cultured gastruloids were mostly composed of mesodermal cells for both replicates tested (S1D–E Fig). We next separated the replicates in the AR and major axis length quantification for pre-cultures 1 and 2. We observed that the TR line remains consistent across replicates, the IB10 line replicates its pattern of a higher AR in pre-culture condition 2 than 1, and the B6 line shows the largest variation between replicates (S1F Fig). We decided to study the effect of all pre-culture conditions in more detail in the B6 cell line to see if other conditions could lead to more consistent results across replicates.

Because the altered pluripotency state of the different pre-culture conditions could have an effect on symmetry breaking during the gastruloid protocol, we subjected all conditions additionally to a delayed chiron pulse from 72 h - 96 h AA, and assessed them at 120h or 144 h AA (S1G–I Fig). We only observed a significant increase of the aspect ratio for pre-culture condition 3 with a 24h delay of the chiron pulse and read-out (S1H–I Fig). Therefore, we subjected condition 3 pre-cultured cells to a 72 h - 96 h AA chiron pulse and read-out at 144 h AA, while for all other pre-culture mESCs we used the default gastruloid protocol. To investigate what causes this difference between pre-cultures, we assessed the time point of Brachyury (T) induction and polarisation between pre-cultures (S1J–K Fig). We found that while condition 1 results in polarised structures at 72h AA, conditions 2 and 3 display a delay in T induction and polarisation. Condition 4 pre-cultured gastruloids display similar symmetry breaking as compared to condition 1, but with a lower overall expression of T. At 96h AA, all pre-culture conditions result in elongated structures with polarised expression of T.

To refine our intermediate pluripotent states we introduced two additional pre-culture conditions before aggregation. mESCs were cultured in ESLIF with a 48-hour (5) or 24-hour (6) 2i pulse followed by a 48-hour (5) or 24-hour (6) ESLIF pulse before aggregation (Fig 1A). When we split the data by experiment, we observed a high degree of heterogeneity between replicates except for conditions 4 and 6 (Fig 1D–E). Gastruloids generated from pre-culture conditions 5 and 6 show the least variation in AR and major axis length across the two replicates (Fig 1D–E).

To conclude, the pre-culture conditions to which mESCs are exposed indeed affect the formation of gastruloids. For the wild-type cell lines tested, variations in mESC pre-culture result in gastruloids with different morphologies and elongation scores (Fig 1). Finally, pre-culture conditions of mESCs affect reproducibility of gastruloid generation, with pre-cultures 5 and 6 generating the most reproducible results for the B6 line, while for TR and IB10 pre-cultures 1 and 2 were tested and found to be reproducible (Figs 1D–E, S1F).

## mESC pre-culture conditions display two distinct transcriptional and epigenetic signatures

Next, we wondered if the observed phenotypic variability during gastruloid formation is already reflected in the mESC state during aggregation. Single-cell RNA sequencing analysis of the mESCs showed a clear segregation between cells from conditions 1, 5, and 6 (ESLIF conditions) on the one hand, and pre-culture conditions 2, 3, and 4 (2i conditions) on the other (Fig 2A). We confirmed that this difference is not due to plate effects (S2A Fig) or other technical aspects, such as the total genes profiled, total counts recovered, or the percentage of mitochondrial reads per cell (S2B–D Fig). In contrast to the observed variability in gastruloid formation, mESCs within each condition were relatively homogenous as they segregated into distinct groups without distinguishable sub-clusters (Fig 2A). Next, we clustered the cells into cell states and determined the top differentially expressed genes for each cluster using the Wilcoxon rank-sum test. Cell state clusters are largely dominated by cells from the same pre-culture condition, with the exception of one cluster shared by pre-cultures 5 and 6, and two outlier clusters (Fig 2B). These two clusters also show a lower S-phase score compared to the other clusters, indicating a low cell division rate (S2E Fig). We classified one of the two outlier clusters as 'Endoderm Progenitors', due to high and specific expression of endoderm-related genes including *Foxa2*, *Sox17*, *Sox7*, *Gata4*, and *Gata6* (S2F Fig) [52]. The second outlier cluster shows expression of mesoderm specification genes, such as *Mesp2*, *Notch1,* and mesodermal allantois marker *Tbx4* [15,19,53], indicative of Mesoderm Progenitors (S2E–F Fig).

Next, we looked at known markers for core, ground state, naive, and primed pluripotency [23,52,54–58]. The majority of the cells display high expression of core pluripotency genes including *Sox2*, *Esrrb*, *Nanog, Dppa5a* and *Tdgf1 (Cripto)* (S2F Fig) [27,54,56,58]. However, we could discern differences between pluripotency states. 2i-grown mESCs (condition 2) show high expression of Ground State Pluripotency genes *Nanog*, *Epha4*, *Mt1*, *Mt2*, and *Tex19.1* [15,56,59], while condition 3 cells exhibit a Naive Pluripotency signature with *Klf4* expression [23,52,56], and mESCs grown in pre-culture conditions 5 and 6 show a Primed Pluripotency signature [27,55,56] with expression of *Tet2, Dusp6, Sox1, Lefty2,* and *Bmp4* (S2F Fig). Condition 4 mESCs resemble Epiblast-like cells with up-regulation of *Igfbp2, Tuba1a*, and *Fos* [23,57,60], and condition 1 mESCs show expression of factors associated with Sonic Hedgehog(*Shh*) signalling, such as *Gbx2*, *Ptch1*, *Sox11*, and *Lefty1* [61–66], indicative of ectodermal priming (S2F Fig) [67–69]. Calculated correlation scores confirmed the difference between pre-culture conditions (S2G Fig) and confirmed Endoderm Progenitors and Mesoderm Progenitors as outlier populations that likely represent a small fraction of differentiated cells (Figs 2B, S2H). To analyse the transcriptional changes underlying this split between the pre-culture conditions in more detail, we performed differential gene analysis between ESLIF conditions, compared to 2i conditions. Top differentially expressed genes include known pluripotency factors as well as epigenetic factors such as *Utf1*, *Jarid2*, and *Phc1* (Fig 2C). To determine whether intermediate states of pre-culture conditions display different transcriptome signatures which could explain their variations in gastruloid formation, we next determined differentially expressed genes within the ESLIF and 2i conditions. Top upregulated genes in the ESLIF control (condition 1) compared to conditions 5 and 6 included *Ptch1, Chd7* and *Foxp1* with enriched GO analysis terms for biological regulation, system development and neurogenesis, confirming priming towards ectoderm (Fig 2D–E). In contrast, top upregulated genes in the 2i control (condition 2) compared to conditions 3 and 4 included pluripotency genes such as *Tex19.1,* and downregulated genes were enriched for the terms multicellular organism development and system development, indicating that conditions 3 and 4 display a later stage of pluripotency than condition 2 (Fig 2F–G).

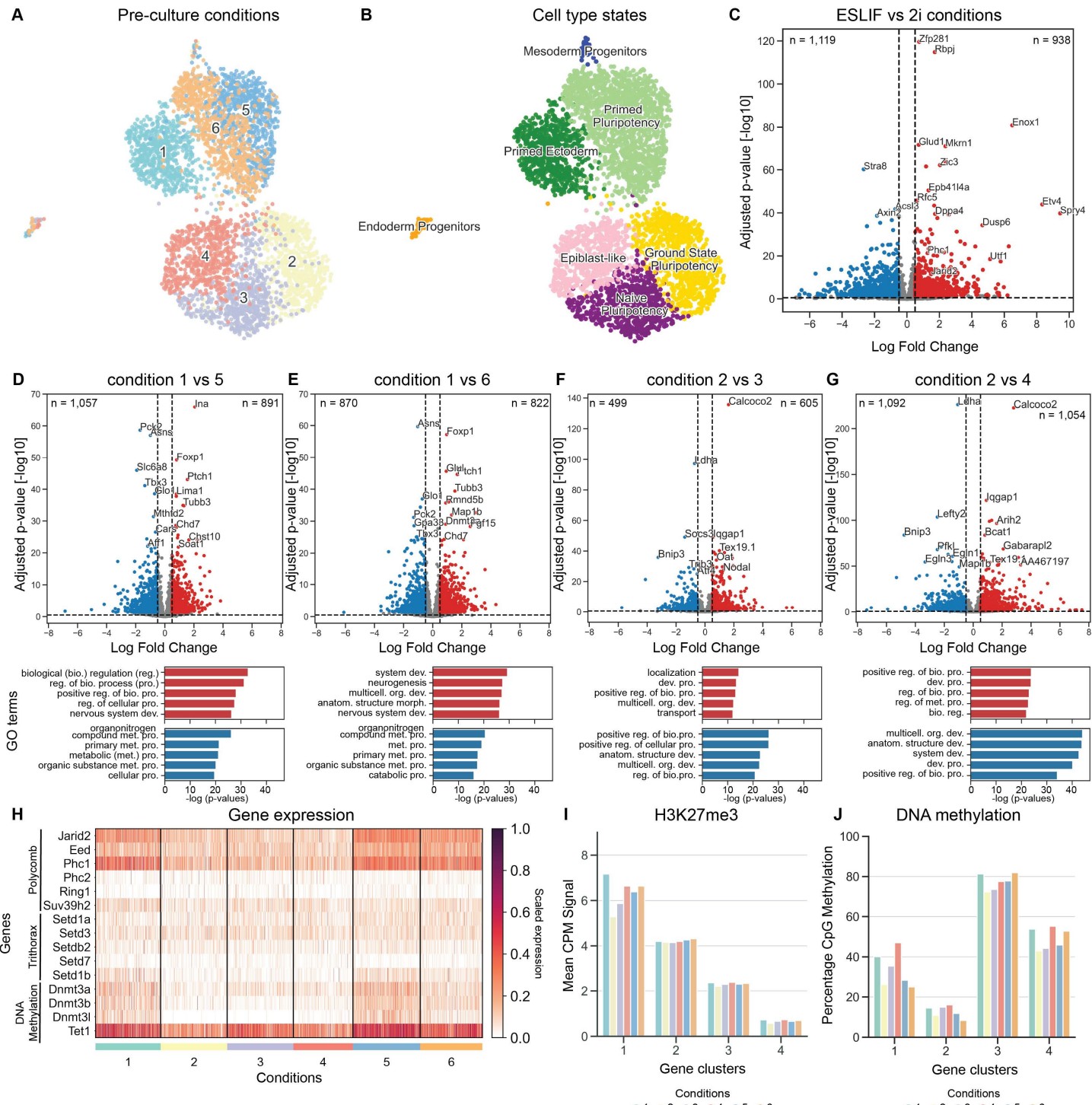

**Fig 2. Specific pulses of culture media result in distinct transcriptome profiles and differential gene expression of epigenetic marker genes in B6 mESCs.** (A) UMAP clustering of single-cell transcriptomes (n = 4,147) of mESCs cultured using pre-culture conditions defined in Fig 1. (B) UMAP coloured by cell type states: Primed Pluripotency (n = 1,396), Ectoderm Priming (n = 658), Mesoderm Progenitorst (n = 28), Endoderm Progenitors (n = 45), Epiblast-like cells (n = 602), cells in Ground State Pluripotency (n = 771), and Naïve Pluripotency (n = 647). Cell typing is based on marker gene expression displayed in S2F Fig and [15,18,19]. (C) Volcano plot indicating downregulated genes (left, blue) and upregulated genes (right, red) in ESLIF pre-culture conditions (conditions 1, 5, 6) compared to 2i pre-culture conditions (conditions 2, 3, 4). (D-G) Volcano plots indicating downregulated genes (left, blue) and upregulated genes (right, red) within ESLIF and 2i pre-culture conditions. The ESLIF control (condition 1) was compared to condition 5 (D) and condition 6 (E), and the 2i control (condition 2) was compared to condition 3 (F) and condition 4 (G). Below: the top 5 GO analysis terms for upregulated (red) and downregulated (blue) genes. (H) Heatmap showing differential expression between the pre-culture conditions of Polycomb-associated genes (*Jarid2, Eed, Phc1, Phc2, Ring1, Suv39h2*), DNA methylation factors (*Dnmt3a, Dnmt3b, Dnmt3l, Tet1*), and

Trithorax-associated genes (*Setd1a*, *Setd3*, *Setdb2*, *Setd7*, *Setd1b*). (I) H3K27me3 abundance in a 20 kb window around the TSS of genes grouped by differential gene clusters. Differential gene clusters were identified based on their H3K27me3 TSS coverage across the six conditions (J) Percentage of DNA methylation on promoter CpGs, grouped per pre-culture condition and gene cluster determined based on H3K27me3 data.

Epigenetic changes between 2i-grown and ESLIF-grown mESCs are well studied, in terms of DNA methylation and chromatin modifications [24–31]. While H3K27me3 is deposited by the Polycomb Repressive Complex (PRC), H3K4me3 is regulated through Trithorax Group Proteins (TxG). Indeed, when we analysed gene expression patterns of Polycomb, Trithorax and DNA methylation factors, we observed an up-regulation of PRC factors in conditions 1, 5, and 6, but not in 2, 3, and 4 (Fig 2H). This same pattern is to a lesser extent replicated in DNA methylation factors, but not in TxG factors (Fig 2H).

To look further into epigenetic differences between the conditions, we profiled DNA methylation with Whole Genome Bisulfite Sequencing (WGBS) and H3K27me3 distribution across the genome with Sort-Assisted Chromatin Immunocleavage (sortChIC) [38]. First, we assessed distribution of both modifications across different genomic regions. While repeat regions are generally highly covered by H3K27me3, gene bodies and promoters contain a lower amount of this mark (S3A Fig). DNA methylation is generally highest in intergenic regions and lowest in promoter regions (S3B Fig). We observed the largest variation between pre-culture conditions in promoter regions, and therefore focused the following analyses on these (S3B Fig). We assessed coverage of H3K27me3 in a 20kb window around individual transcription start sites (TSS) and identified four gene clusters with variable H3K27me3 distribution dynamics (Fig 2I). To assess the effects of these epigenetic differences on gene expression, we grouped the scRNA-seq dataset generated before according to the same gene clusters and analysed the average expression per pre-culture condition and gene cluster (S3C Fig). Around 20% of the genes classified to clusters 1 and 2 are differentially expressed between the ESLIF and 2i conditions, with more genes being downregulated (14.5% and 12.4%, resp.) than upregulated (8.3% and 9.4%, resp., S2E Fig).

Generally, gene cluster 1 shows variability in the amount of DNA methylation and H3K27me3 coverage between pre-culture conditions, but not when assessing gene expression (Figs 2I–J, S3C). GO analysis reveals that the variable gene cluster 1 is associated with terms like system development, multicellular organism development, and cell differentiation (S3D–E Fig). In contrast, while genes classified in cluster 2 show generally high levels of H3K27me3 coverage, DNA methylation across these gene promoters is low with levels below 20% (Fig 2I–J). Genes classified to gene clusters 1 and 2 are generally lowly expressed, consistent with their silencing through H3K27me3 (Figs 2I–J, S3C). Genes classified to cluster 3 follow the inverse pattern with DNA methylation levels reaching 80%, intermediate levels of expression and corresponding lack of H3K27me3. Genes classified to cluster 4 show lower levels of both DNA methylation and H3K27me3 coverage, consistent with generally high levels of gene expression (Figs 2I–J, S3C). Lastly, we asked whether differentially expressed genes undergo epigenetic changes. For this, we compared the gene-wise H3K27me3 promoter abundance in the different pre-culture conditions and labelled differentially expressed genes (S3F Fig). This showed a clear loss of H3K27me3 on transcriptionally upregulated genes of the same condition. Importantly, differentially expressed genes show overall relatively low H3K27me3 levels in all pre-culture conditions, which get further reduced during gene activation.

## mESC pre-culture conditions affect lineage contribution in gastruloids

To study the effect of different mESC pre-culture conditions on the differentiation potential, we generated gastruloids from pre-culture conditions 1, 5, and 6 as described above.

Pre-culture condition 1 was selected as a control, and conditions 5 and 6 were selected because they gave the most consistent results with higher AR scores in previous assays (Figs 1–2, S1–S3). Considering their similar mESC profile and gastruloid morphologies upon differentiation, we hypothesised that conditions 5 and 6 would also yield comparable cell type compositions, while condition 1 might show a bias towards ectoderm consistent with its priming through Shh signalling in mESCs (S1A, 2B, S2F Figs). For each pre-culture condition, we selected 5 single gastruloids and subjected them to single-cell RNA sequencing to evaluate reproducibility. After excluding low-quality cells (see Methods), the remaining 4,637 single cells were used for UMAP embedding (Figs 3A, S4A). Gastruloids from every condition contain cell types derived from all three germ layers, with mesoderm contributing the highest and endoderm contributing the lowest number of cells (Figs 3B–C, S4B). Quality metrics including the total number of genes profiled, total counts recovered, and percentage of mitochondrial reads per cell are comparable between conditions and cell types (S4C Fig). Cell types were annotated based on established marker genes shown in [14,15,18,19].

When we assessed the ratio of cell types for each gastruloid derived from each condition, we observed that condition 1 is skewed towards an ectodermal fate and generates a significantly higher ratio of Spinal Cord cells (Figs 3D–E, S4D–F). This is consistent with ectoderm priming in condition 1 mESCs, identified by upregulation of Shh signalling factors (Figs 2B, S2F). In contrast, gastruloids derived from conditions 5 and 6 generate a significantly higher fraction of mesoderm cells, with a bias towards Somites and Early Somites, respectively (Figs 3D–E, S4D–F). Lastly, while condition 1 gastruloids produce the rare ectodermal cell type Neural Tube Progenitors, condition 5 and 6 gastruloids generate Endothelium (S4E–F Fig). Of note, these results are consistent with the FACS observations from gastruloids generated from the TR cell line (S1D Fig).

To discard a delay in differentiation as a cause for differential lineage contribution, we integrated this dataset with published data profiling gastruloids from 72h until 168h AA with 24h intervals [44]. We chose this dataset because it was generated using the same cell line, and gastruloids were generated from mESCs grown according to pre-culture condition 5. Upon integration, gastruloids from pre-culture conditions 1, 5, and 6 mainly corresponded to the expected 120h AA sampling time point (Figs 3F, S4G–H) and displayed a similar distribution across pseudotime (Fig 3G). In contrast, when comparing the lineage contributions defined by the reference dataset, the bias of condition 1 gastruloids towards ectoderm and that of conditions 5 and 6 towards mesoderm was confirmed (Fig 3H). Observing these clear differences in differentiation, we set out to determine whether the epigenetic state of the pre-culture mESCs already gives an indication of the observed bias. To evaluate this, we determined differentially expressed genes in the three germ layers based on the reference dataset [44]. Plotting the H3K27me3 count distributions over these gene sets in mESCs showed different enrichment between gene sets and conditions (S4I Fig). While endodermal genes are low on H3K27me3 in all 3 pre-culture conditions, both ectodermal and mesodermal genes show overall lower H3K27me3 levels in condition 1 relative to condition 5 and 6. Lastly, pre-culture conditions display comparable expression of Hox genes across different germ layers and pre-culture conditions (S4J Fig).

## Discussion

Here, we show that by modulating the mESC state through pre-culture media composition, the elongation length and reproducibility of gastruloid formation is affected. We first tested three different cell lines of various genetic backgrounds and noticed that the pre-culture condition resulting in the highest AR differs per cell line, highlighting the need for cell line-specific optimisation of the gastruloid protocol. While the B6 background yielded the lowest

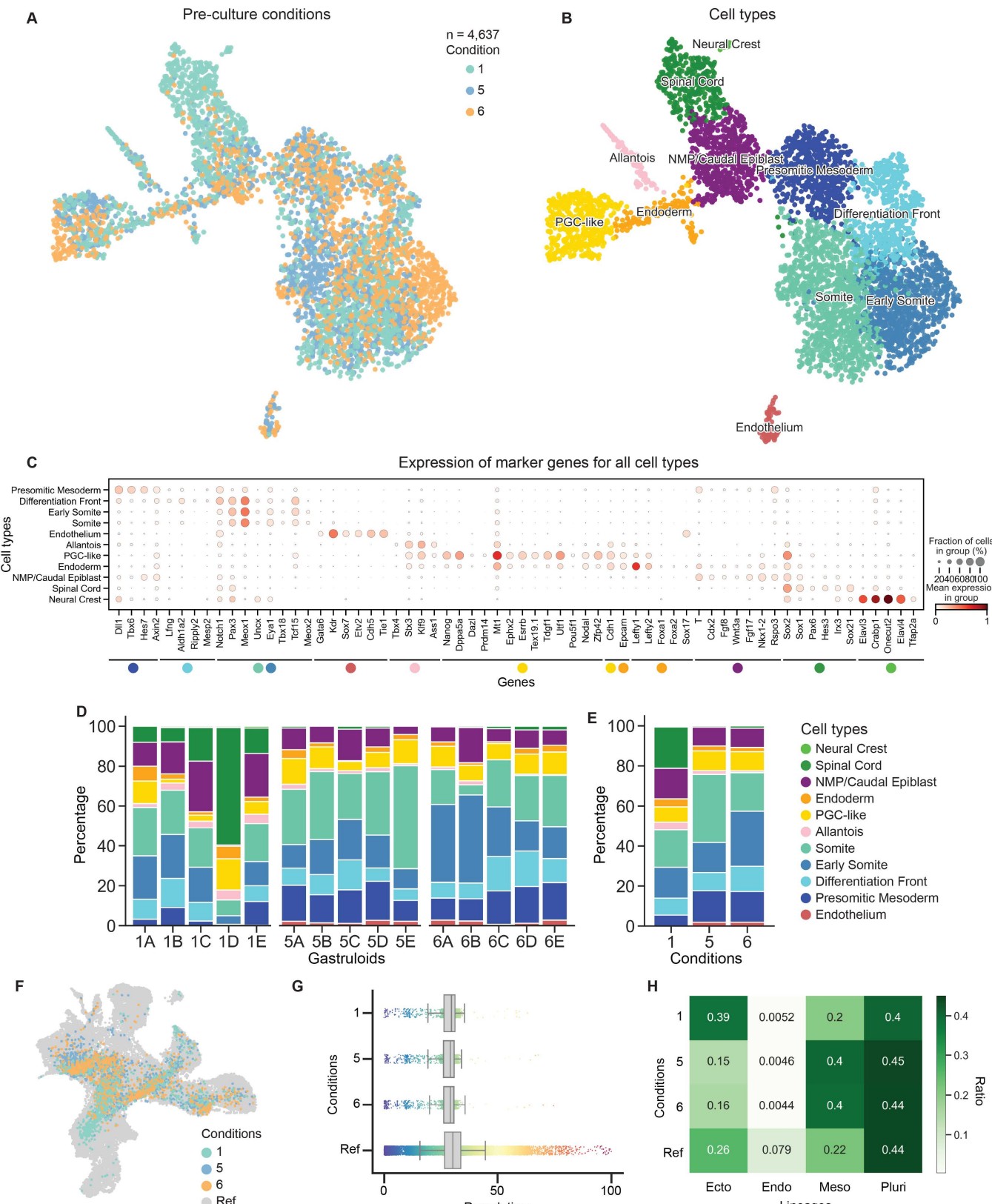

**Fig 3. Pre-culture conditions affect lineage contribution in gastruloids generated with B6 mESCs.** (A) UMAP showing clustering of single-cell RNA sequencing data of five individual gastruloids for conditions 1 (n = 1,529), 5 (n = 1,532), and 6 (n = 1,576), sampling a total of 4,637 cells. (B) UMAP

coloured by recovered cell types: Primordial Germ Cell-like Cells (PGC-like, n = 419); Allantois cells (n = 96); Endodermal cells (n = 134); mesodermal cell types, including Early Somites (n = 903), Somites (n = 1,111), Presomitic Mesoderm (n = 563), and Differentiation Frontt (n = 467), Endothelium (n = 62), Caudal Epiblast containing NMPs (n = 529); and ectodermal cell types including Spinal Cord (n = 344), and Neural Crest (n = 9). Cell type annotation is based on localised expression of established marker genes (see Fig C) [14,15,18,19]. (C) Dot plot showing the expression of marker genes in all cell types. Dot colour represents the mean expression of marker genes in specific groups, and dot size represents the fraction of cells within the group that show expression of the gene. (D) Cell type distributions across 5 individually sampled gastruloids from condition 1 (1A-E), condition 5 (5A-E), and condition 6 (6A-E). (E) Average cell type distributions from (D) between conditions 1, 5, and 6. (F) Integration of gastruloid preculture dataset with gastruloid time-course dataset from 72h AA until 168h AA [44]. (G) Pseudotime distributions of integrated dataset split into preculture conditions 1, 5, and 6, and reference dataset. (H) Ratios of lineage contributions of integrated dataset split into preculture conditions 1, 5, and 6, and reference dataset.

AR of gastruloids after 2i culture, this was not replicated in the TR and IB10 cell lines. In addition, another study using E14 mESCs reported increased efficiency of elongation after mESCs were maintained in 2i medium, and some cell lines require a 2i pulse for gastruloid formation [51,54]. It is therefore plausible that different cell lines require alternative pre-treatments to achieve the same receptive state as required for symmetry breaking during gastruloid formation. Additional experimental variations between studies, such as alterations in media composition and variations in the cell culture protocol, could also influence the reported differences. In addition, we observe that shifting the time point of the Wnt activation pulse could improve gastruloid elongation depending on the pre-culture condition used, and propose to take this into consideration when performing the gastruloid protocol with untested cell lines.

Next, we suggest that modulating the pluripotency state can also affect downstream lineage potential upon differentiation, consistent with current literature [70]. Concerning gastruloid differentiation, a study reporting mESC maintenance in ESLIF medium identified a high ratio of ectoderm cells, while another study routinely maintained mESCs in ESLIF supplemented with 2i and reported faithful emergence of endoderm cells [15,20]. Since the impact of such long term 2i treatment on ES cell differentiation state and gastruloid formation was already thoroughly studied, in this work we focused on shorter 2i pulses to adjust and synchronise their cellular states. The gastruloid protocol was also reported to result in inter-gastruloid heterogeneity, with individual gastruloids being biased towards either an ectoderm or a mesoderm fate [22]. This bias in specification was caused by differences in response strength to the Wnt-activation pulse. To take inter-gastruloid variability into account, we profiled 5 individual gastruloids per pre-culture and compared lineage contributions between replicates and between conditions. We did not observe similar inter-gastruloid heterogeneity in this dataset. Future studies are required to shed further light on how cellular heterogeneity in responsiveness to extracellular signals contributes to lineage decisions. In our study, synchronising mESCs with a 2i-pulse before exposing them to ESLIF followed by aggregation yielded a higher fraction of mesoderm compared to the ESLIF-only control. Maintenance of mESCs in ESLIF produced a higher ratio of ectodermal cells in gastruloids, consistent with ectodermal priming through Shh signalling observed under these conditions. This indicates that a higher ratio of ectoderm cells upon differentiation is already visible by early activation of Shh signalling in mESCs, and can be identified by targeted screening for the differentially expressed genes. However, overall differences between mESCs in the pre-culture conditions tested are low, both in their transcriptome and epigenetic profiles. Generally, mESC pre-cultures that generated more consistent gastruloids in the B6 cell line were marked by upregulation of epigenetic regulators of DNA methylation and H3K27me3, corresponding with differential coverage of these marks across promoter regions of developmental regulators. However, this signature did not explain the difference in cell fate composition between pre-culture 1 and pre-cultures 5 and 6. In addition, genes activated in the ectoderm and mesoderm lineages during gastruloid formation display reduced coverage by H3K27me3 in condition 1 compared

to condition 5 and 6 pre-cultured mESCs. It remains to be determined why higher H3K27me3 coverage on both gene sets favours mesodermal differentiation, while lower H3K27me3 levels promote ectodermal differentiation. Future work looking at multiple states along differentiation might show clearer and more predictive differences.

Importantly, building upon previous work to systematically improve gastruloid efficiency and complexity [71,72], we present an unbiased screening approach to assess the efficiency of gastruloid formation. Through imaging of whole gastruloid plates and the quantification of morphological properties including aspect ratio and absolute measures of the major and minor axis length, the effect of different treatments can be easily assessed. Selected conditions are then further characterised in detail to identify the impact on ES cell state and gastuloid cell type composition. Using this approach, we envision that gastruloid culture can be systematically optimised for reproducibility in a wide variety of cell lines and genetic backgrounds, before a more detailed analysis of the resulting cell composition is assessed.

## Supporting information

**S1 Fig. Variations in mESC pluripotency state affects gastruloid morphology, perceptiveness to chiron pulse, elongation length and efficiency of symmetry breaking** . (A) Bright-field images showing mESC morphologies of condition 1 (serum) and condition 2 (2i) pre-cultures for all three cell lines. (B) Examples of gastruloid morphologies generated from the different pre-culture conditions for the three different cell lines. (C) Workflow for quantification of major axis length, minor axis length and aspect ratio of gastruloids. 96-well plates with gastruloids were imaged, and images of gastruloids were loaded into Fiji. Images were segmented using Ilastik, MOrgAna [36] was used to calculate the major and minor axis length. The aspect ratio was calculated by dividing the major axis length by the minor axis length of each gastruloid. (D-E) FACS plots (D) and quantification (E) of gastruloids formed using the TR line for condition 1 (serum) and condition 2 (2i). Mt1:BFP marks ectoderm (ect), T:GFP mesoderm (mes), and Sox17:RFP endoderm (end). (F) Aspect ratio and major axis length of gastruloids after condition 1 (serum) and condition 2 (2i) pre-cultures from all three cell lines, separated by replicate. Statistical significance between replicates A and B was calculated with an independent t-test. (G) Schematic overview of chiron pulse and read-out optimisation. For each pre-culture condition, the effect of conventional timing of the chiron pulse (48-72h AA: light orange, dark orange) and a delayed chiron pulse (72-96h AA: light purple, dark purple) on gastruloid formation were assessed. Gastruloid formation was assessed at 120 h AA and 144 h, and the aspect ratio (H) and major axis length (I) were calculated. Statistical significance of the differences in chiron pulse and read-out within conditions were calculated with Mann-Whitney U test with Bonferroni multiple testing correction. (J) Sox2 (green) and T (magenta) stainings and quantification (K) of pre-cultured gastruloids at 48h, 72h, and 96h AA. ES cell cultures were pre-treated with ESLIF (c1) or 2i from 96h (c2), 48h (c3) or 24h (c4) prior to the start of the gastruloid protocol. Depicted are maximum projections of representative gastruloids. In the right bottom corner, the proportion of representative gastruloids over the total sample size is indicated (n = 2, with the exception of C3-96h AA (n = 1)). White arrowheads point to hard-to-see positive cells. Scale bar = 100µm. Images were obtained using a Leica SP8 confocal microscope.
(TIF)

**S2 Fig. Specific pulses of mESC culture media result in distinct transcriptome profiles in B6 mESCs.** (A) UMAP coloured by plate distribution, insert: ratios of cell type states per plate. (B-D) Distribution of total count distribution (B), mitochondrial gene expression (C), and total gene count distribution (D) per cell. (E) S-phase scores per cell type. (F) Dot plot

showing expression of marker genes used for determination of cell type states displayed in Fig 2B. Dot colour represents the mean expression of marker genes in specific groups, and dot size represents the fraction of cells within the group that show expression of the gene. (G-H) Correlation maps between conditions (G) and between cell type states (H).
(TIF)

**S3 Fig. Specific pulses of culture media result in distinct epigenetic profiles in B6 mESCs. (A)** Normalised H3K27me3 abundance across different genomic regions for all six mESC pre-culture conditions tested. Counts are normalised for features per million (FPM), which takes into account the variable sequencing depth across samples and the variation in genome size of the different features. (B) Percentage of CpG methylation across different genomic regions for all six mESC pre-culture conditions tested. (C) Average level of transcriptome reads derived from scRNAseq dataset, grouped per pre-culture condition and gene cluster determined based on H3K27me3 data. (D) Volcano plots corresponding to Fig2C separated by identified gene clusters, indicating down regulated genes (left, blue) and upregulated genes (right, red) in ESLIF pre-culture conditions (conditions 1, 5, 6) compared to 2i pre-culture conditions (conditions 2, 3, 4). (E) Top 10 GO terms associated with gene cluster 1. (F) Scatterplots showing H3K27me3 counts (CPM normalised), found in 15 kb TSS windows comparing different pre-culture conditions. Coloured dots represent genes differentially expressed between indicated conditions, with red indicating genes upregulated and blue downregulated in the condition shown on the x-axis. Coloured numbers in the corners indicate the fraction of differentially expressed genes showing reduced H3K27me3 in the indicated condition.
(TIF)

**S4 Fig. Variations in pluripotency state affect cell type composition of gastruloids generated with B6 mESCs.** (A) Distribution of replicates across transcriptome UMAP. Five individual gastruloids were included for pre-culture conditions 1, 5, and 6. (B) Transcriptome UMAP separated by germ layers - endoderm (n = 230), mesoderm (n = 3,106), ectoderm (n = 882) - and pluripotency cells (n = 419). (C) Distribution of mitochondrial gene expression (left), total count distribution (middle), and total gene count distribution (right) per cell. (D) Ratio of germ layer contributions across 15 individually sampled gastruloids from condition 1 (1A-E), condition 5 (5A-E), and condition 6 (6A-E) (left) and between conditions 1, 5, and 6 (right). (E-F) Boxplots of cell type (E) and lineage (F) percentages per condition. Statistical significance between conditions was calculated with Wilcoxon rank-sum test with Bonferroni multiple testing correction. (G) Density plots of pre-culture dataset separated into conditions 1, 5, and 6, and the reference dataset. (H) Ratios of sampling times mapped onto the pre-culture dataset separated into conditions 1, 5, and 6, and the reference dataset. (I) Histograms showing H3K27me3 counts per TSS (CPM normalised), split by group of differentially expressed genes in gastruloids. (J) Heatmap of Hox gene expression across germ lines and conditions.
(TIF)

## Acknowledgments

The authors thank A. van Oudenaarden for supporting this study. We thank V. van Batenburg for help with the imaging experiments and for useful discussions. J. Verity-Legg assisted with bisulfite sequencing experiments. We are grateful to S. van den Brink and S. de Vries for sharing their cell culture expertise, and to D. Turner for providing useful information and discussions. We thank the Hubrecht Sorting Facility (HSP) and Utrecht Sequencing Facility (USEQ) for cell sorting and sequencing. We thank the ENW XL consortium "Stochasticity" for their

suggestions on the cell lines and conditions. We also thank R. van Amerongen for help with revisions. We thank the laboratory of Iftach Nachman (Tel Aviv University) for sharing the dual BRA/SOX17 reporter cell line, and Suzan Stelloo for sharing the triple reporter line as generated from this double reporter. The 129S1/SvImJ/ C57BL/6 mESC line was a gift from Matyas Flemr and Marc Bühler.

## Author contributions

**Conceptualization:** Marloes Blotenburg.

**Data curation:** Marloes Blotenburg, Lianne Suurenbroek, Danique Bax, Joelle de Visser.

**Formal analysis:** Marloes Blotenburg, Lianne Suurenbroek, Danique Bax, Joelle de Visser, Vivek Bhardwaj.

**Funding acquisition:** Hendrik Marks.

**Investigation:** Marloes Blotenburg, Lianne Suurenbroek, Danique Bax, Joelle de Visser.

**Methodology:** Marloes Blotenburg, Lianne Suurenbroek, Danique Bax, Joelle de Visser.

**Project administration:** Hendrik Marks.

**Resources:** Luca Braccioli, Elzo de Wit.

**Supervision:** Marloes Blotenburg, Elzo de Wit, Antonius van Boxtel, Hendrik Marks, Peter Zeller.

**Validation:** Marloes Blotenburg, Lianne Suurenbroek, Danique Bax, Joelle de Visser, Hendrik Marks, Peter Zeller.

**Visualization:** Marloes Blotenburg, Peter Zeller.

**Writing – original draft:** Marloes Blotenburg, Lianne Suurenbroek, Peter Zeller.

**Writing – review & editing:** Marloes Blotenburg, Lianne Suurenbroek, Danique Bax, Antonius van Boxtel, Hendrik Marks, Peter Zeller.

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
