## [Decision Letter · Decision Letter 0]

18 Jun 2024

PONE-D-24-20935Stem cell culture conditions affect in vitro differentiation potential and efficiency of mouse gastruloid developmentPLOS ONE

Dear Dr. Blotenburg,

Thank you for submitting your manuscript to PLOS ONE. After careful consideration, we feel that it has merit but does not fully meet PLOS ONE’s publication criteria as it currently stands. Therefore, we invite you to submit a revised version of the manuscript that addresses the points raised during the review process.

You should try to address all of the Reviewers' points detailed below, through experiments or by added detail, clarification or discussion.It is particularly important that you establish the reproducibility of your findings through analysis of at least one additional mouse ES cell line, from a distinct genetic background. It is also critical that you include detailed experimental descriptions, including considerations of cell density, molar concentrations of growth factors, sources of media and additives, as requested by the Reviewers. Also provide more detail in terms of the design of your scRNA-seq experiments. Please include time-dependent analysis of key regulators during gastruloid formation, e.g. by immunostaining or ISH, as a means of linking lineage priming bias in 2D cultures and differential lineage representation in the gastruloids.  

We look forward to receiving your revised manuscript.

Kind regards,

Cristina Pina, MD DPhil

Academic Editor

PLOS ONE

Journal Requirements:

"This work was supported by a European Research Council Advanced grant (https://erc.europa.eu/, ERC-AdG 742225-IntScOmics) and a Nederlandse Organisatie voor Wetenschappelijk Onderzoek (NWO) TOP award (https://www.nwo.nl/, NWO-CW 714.016.001). P.Z. was funded by the Swiss National Science Foundation (https://www.snf.ch/, SNF, P2BSP3-174991), Human Frontier Science Program (HFSP, https://www.hfsp.org/, LT000209/2018-L) and Marie Skłodowska-Curie Actions (https://marie-sklodowska-curie-actions.ec.europa.eu/, 798573). V.B. was funded by European Molecular Biology Organization Long-term Fellowship (https://www.embo.org/, EMBO LTF, ALTF 1197–2019). This work is part of the Oncode Institute which is partly financed by the KWF Dutch Cancer Society (https://www.oncode.nl/). The funders did not play a role in the study design, data collection and analysis, decision to publish, or preparation of the manuscript."

"The authors thank A. van Oudenaarden for supporting this study. We thank V. van Batenburg for help with the imaging experiments and for useful discussions. J. Verity-Legg assisted with bisulfite sequencing experiments. We are grateful to S. van den Brink and S. de Vries for sharing their cell culture expertise, and to D. Turner and T. van Boxtel for providing useful information and discussions. We thank the Hubrecht Sorting Facility (HSP) and Utrecht Sequencing Facility (USEQ) for cell sorting and sequencing. The 129S1/SvImJ / C57BL/6 mESC line was a gift from Matyas Flemr and Marc Bühler. This work was supported by a European Research Council Advanced grant (ERC-AdG 742225-IntScOmics) and a Nederlandse Organisatie voor Wetenschappelijk Onderzoek (NWO) TOP award (NWO-CW 714.016.001). P.Z. was funded by SNF (P2BSP3-174991), HFSP (LT000209/2018-L) and Marie Skłodowska-Curie Actions (798573). V.B. was funded by EMBO LTF (ALTF 1197–2019). This work is part of the Oncode Institute which is partly financed by the KWF Dutch Cancer Society. "

"This work was supported by a European Research Council Advanced grant (https://erc.europa.eu/, ERC-AdG 742225-IntScOmics) and a Nederlandse Organisatie voor Wetenschappelijk Onderzoek (NWO) TOP award (https://www.nwo.nl/, NWO-CW 714.016.001). P.Z. was funded by the Swiss National Science Foundation (https://www.snf.ch/, SNF, P2BSP3-174991), Human Frontier Science Program (HFSP, https://www.hfsp.org/, LT000209/2018-L) and Marie Skłodowska-Curie Actions (https://marie-sklodowska-curie-actions.ec.europa.eu/, 798573). V.B. was funded by European Molecular Biology Organization Long-term Fellowship (https://www.embo.org/, EMBO LTF, ALTF 1197–2019). This work is part of the Oncode Institute which is partly financed by the KWF Dutch Cancer Society (https://www.oncode.nl/). The funders did not play a role in the study design, data collection and analysis, decision to publish, or preparation of the manuscript."

4. "n the online submission form, you indicated that [The GEO submission of sequencing data is unfortunately delayed through the holidays, but we will provide the accession number and access link as soon as we receive them. Until then, all data is available upon request.]. 

Reviewers' comments:

Reviewer's Responses to Questions

**Comments to the Author**

1. Is the manuscript technically sound, and do the data support the conclusions?

Reviewer #1: Partly

Reviewer #2: Partly

2. Has the statistical analysis been performed appropriately and rigorously?

Reviewer #1: Yes

Reviewer #2: Yes

3. Have the authors made all data underlying the findings in their manuscript fully available?

Reviewer #1: No

Reviewer #2: Yes

4. Is the manuscript presented in an intelligible fashion and written in standard English?

Reviewer #1: Yes

Reviewer #2: Yes

5. Review Comments to the Author

Reviewer #1: In the manuscript entitled “Stem cell culture conditions affect in vitro differentiation potential and efficiency of mouse gastruloid development” Blotenburg and colleagues confirmed and extended previous observations that the cell culture conditions, and specifically the pluripotency state of the stem cell population, affect gastruloid development in term of morphology, elongation, and cell type composition. By performing ScRNA-seq analysis of mESCs and gastruloids the authors identify the pathways that control the efficiency of gastruloid formation. The results are interesting and provide insights into the complexity of stem cell plasticity. However, in my opinion, there are important points that need to be addressed prior to publication.

Major points:

The impact of different culture conditions on gastruloids formation have been previously reported. Unfortunately, the authors incorrectly cite the relevant literature. For instance, Cermola et al (Stem Cell Report 2021, ref.31) reported the effect of ESCs grown in different culture conditions//pluripotency state including but not limited to i) DMEM/FBS 15%/LIF, ii) N2B27 2i/LIF, and FGF/Activin -induced EpiLC and EpiSCs on gastruloids formation. They demonstrated that culture conditions significantly impact gastruloid formation, affecting i) cell aggregation (size of cell aggregates at 48 hours), ii) elongation (proportion of correctly elongated gastruloids), and iii) morphology (presence of aberrant gastruloids/multiple protrusions).

The authors should discuss their findings more thoroughly in the context of the current literature and properly cite the relevant papers. This will not diminish the significance of their studies.

Materials and Methods

- Pag. 4, line 84: PBS0 and TrypLE are not defined. Please specify

- Pag. 5, lines 92-93: the concentration of 2i inhibitors (CHIR and PD) are expressed as %, but it is not clear % of what mass/vol?. Please clarify or indicate the molar concentration.

- Pag. 7, line 147: the sentence ‘Cells with either less than 500 reads and 100 genes detected in less than 2 cells were filtered out’. Please rephrase or clarify

Results

Pag. 8 lines 195-196: the authors claim that ..”pluripotency states in between naïve and ground state can be produced with short-term pulses of 2i and ESLIF”. However, this conclusion lacks experimental evidence. The authors should provide at least some characterization of the cells defined as “intermediate” states of pluripotency (conditions named 3, 4, 5 and 6) between conditions 1 and 2.

Figure 1, pre-culture conditions, including the number of cells plated/density, the type of plates and coating used, are not specified.

Figure 1, panel A’. At the high cell density, assessing the effect of culture conditions on cell colony morphology seems difficult to evaluate/quantify. Therefore, the phenotype-based analysis of the intermediate states proposed here lacks convincing evidence. On the other hand, a molecular-based approach, i.e. the identification of specific markers for the 6 conditions, for instance based on data from the RNAseq analysis, seems not sufficient per se (see below).

Pag. 9 line 204. …”cells from all pre-culture conditions except ‘condition 2’ generate gastruloids efficiently”. The 2i medium is routinely used to capture naïve pluripotency, and so, the observation that ESCs grown in 2i medium (condition 2) are unable to elongate is unexpected. It is likely that 2i ESCs are ‘locked” in the naïve pluripotent state and that this eventually delay/prevent exit from pluripotency and symmetry breaking. Indeed, condition 5, that is 48h in LIF medium is sufficient to rescue the elongation efficiency (see also Discussion section below). Please, discuss.

Figure 1, the values of the elongation index are overall lower than those previously reported by different laboratories (2.5 or higher), and the abnormal value (see Fig.1, panel D, conditions 3, 4, 6 replicate B) appears to correspond to the diameter rather than the length. Indeed, the diameter is larger than expected (100-200 microM). Likely, it is due to the use of the largest circle (from Max Inscribed Circles Fiji plug-in) as the gastruloid diameter.

Pag. 10 line 242-243, ScRNAseq analysis (Figure 2) revealed that ESCs even if grown in 6 different conditions, display only two different transcriptional and epigenetic signatures. These results raised some issues:

a.- What about the intermediates states of pluripotency. Is the ScRNAseq analysis able to discriminate the cells grown in the 6 conditions? The cells pre-grown in the 6 different conditions display significant differences in their transcriptome? Please clarify

b. Is the transcriptome profile relevant for the gastruloid formation efficiency?. For instance, cells from conditions 1, 5, and 6 seems to segregate together from the seq analysis but display a completely different gastruloid formation efficiency (Figure 1).

It seems that the presence of LIF for the last 24 hour (compare conditions 1, 5, and 6) is sufficient to induce a similar signature/cell identity (transcriptome and epigenetic signature).

Pag. 13 line 323 to Pag. 14 line 346, ScRNAseq analysis (Figure 3) revealed that gastruloids generated with cells grown under different conditions (1, 5, and 6), have a different lineage compositions. These interesting results raise some issues:

a.- Is the composition of the gastruloids analyzed here (mainly condition in line with that reported in the literature with respect to types/number of cell lines identified, etc.?

b.- the high fraction of mesoderm in gastruloids derived from conditions 5 and 6 correlates with the induction of mesodermal markers in ESCs grown under these conditions?

Discussion

Pag. 14, lines 362-364. …”While in our study 2i grown mESCs yielded the lowest efficiency of gastruloid elongation, another study using E14 mESCs reported increased efficiency of elongation after mESCs were maintained in 2i medium. (31)”. This is not correct. In the paper by Cermola et al. (ref 31 ms), mESCs were maintained (passaged) in DMEM/FBS 15%/LIF not in 2i. Then, to generate gastruloids, mESCs were seeded at a low density (300 cells/cm²) on gelatin-coated plates and cultured in N2B27 2i/LIF for 5 days. It is important to note that the 2i medium also includes LIF (2iLIF). Under these culture conditions, 90%–95% of ESC colonies show a domed round-domed shaped phenotype. Besides the difference in the culture conditions, other technical differences in the protocol need to be consider:

1) mESCs were dissociated with a milder accutase treatment, and not trypsin

2) The protocol includes a FACS sorting step to eliminate dead cells and cellular debris prior to aggregation.

Thus, the culture conditions and the protocols used in the paper by Cermola et al. (ref. 31) and those reported herein are different and should be considered. These important differences should be addressed and discussed, and the sentence should be rephrased accordingly.

Reviewer #2: The study of Blotenburg et al. addresses whether stem cell culture conditions, particularly the introduction of 2i, affect the development of mouse gastruloids and whether an optimised mESC pluripotency state would provide more consistent results. These are relevant questions because the gastruloid model system is widely used and, contrary to the 3D/N2B27 part, 2D culture conditions lack a defined protocol and have been shown to vary substantially (sometimes even in the same study).

Throughout the manuscript the authors present results suggesting that modulation of the pluripotency state of mESCs significantly affects the phenotype (morphology), transcriptome and epigenome of gastruloids, ultimately impacting the specification of certain cell fates. Although these results are interesting, they were obtained from only one mouse cell line. Also, the same occurs for the optimisation work carried out to improve the reproducibility/consistency of the gastruloid system. Checking the ‘universality’ of their findings in at least two more mESCs (also used by other labs working with gastruloids) would significantly improve the work and its significance to the community. Another aspect that requires further work regards the control of cell number during 2D culture. According to the manuscript, cells were split every second day in a 1:5 ratio during the modulation of their pluripotency state. Did the authors examine whether cell number was kept stable across the different conditions? If the numbers changed, they should check whether seeding density alone affects the pluripotency state and, further down, gastruloid formation. It is important to isolate/control the seeding density variable during these experiments to understand the 2i effects more accurately.

After modulating the 2D culture conditions, Blotenburg et al. developed mouse gastruloids according to the standard protocol and performed a detailed morphological analysis. Here, it would be nice if the authors pointed out the morphological differences they see in the gastruloid images of Figure S1A (e.g. “ridge in the centre and protrusions at the posterior end”). The authors then tested the effects of a temporal shift in the CHIR treatment in the various culture conditions. However, it is not possible to correctly interpret this experiment without any Brachyury stainings at 48, 72 and 96h. Also, Blotenburg et al. should perform the read-out of condition 2 at 144h; there is published data indicating that gastruloids whose cells have been treated with 2i only achieve their maximum elongation at that time (e.g. Beccari et al., 2018). Finally, the authors should also revise terms like “ ‘perceptiveness’ to the CHIR treatment” and explain the meaning of “ ‘efficient’ gastruloid formation”.

In the last part of the manuscript, the authors address the transcriptome and epigenome comparison they did for the different culture conditions, both before and post gastruloid formation, and conclude that there are significant differences in their molecular signatures. Here the authors must revise their annotations and try to be more accurate in defining the different pluripotency states and cell types present in both the pre-culture and gastruloid datasets. For defining the different pluripotent cell states, the authors should use terminology, supported by gene references, that have already been published in other studies (e.g. Nichols and Smith, 2009, Smith 2017, Morgani et al., 2017 and Cermola et al., 2021). Regarding the gastruloid data, Blotenburg et al. should expand the number of genes used for annotation (e.g. Tbx1, Tcf15, Sox17, Noto, Pax6, Otx2, etc.) and use correct anatomical/embryological terminology (e.g. ‘Caudal epiblast’ instead of ‘Tailbud’, ‘Spinal cord’ instead of ‘Neurons’, etc.). Importantly, the authors also need to look at genes that can provide some information about developmental time as it is critical to discard heterochrony; UMAPs showing the integration of the different gastruloid datasets with that of embryos would be very helpful as well. Additionally, the authors should include in the materials and methods section information about the number of cells analysed per condition during the previous single cell experiments.

One of the key results of this manuscript is the finding that modulation of pre-culture conditions, with the addition of 2i, seems to have a strong effect later on cell fate specification during gastruloid formation, particularly in the amount of neural and mesodermal tissues. This finding contrasts with what is reported in the preprint of Rosen et al., 2022, which also finds a bias in the amount of neural and mesodermal cells but links it to inter-gastruloid heterogeneity; Blotenburg et al. should discuss this issue.

Overall, the main question addressed in this manuscript is pertinent and the experiments are interesting. However, there is a considerable amount of work to be done in order for the results to be accurate and significant to the community.

6. PLOS authors have the option to publish the peer review history of their article (what does this mean? ). If published, this will include your full peer review and any attached files.

**Do you want your identity to be public for this peer review?** For information about this choice, including consent withdrawal, please see our Privacy Policy .

Reviewer #1: No

Reviewer #2: No

---

## [Author Response · Author response to Decision Letter 1]

21 Aug 2024

Point-by-point response to the reviewer comments:

Reviewer #1: In the manuscript entitled “Stem cell culture conditions affect in vitro differentiation potential and efficiency of mouse gastruloid development” Blotenburg and colleagues confirmed and extended previous observations that the cell culture conditions, and specifically the pluripotency state of the stem cell population, affect gastruloid development in term of morphology, elongation, and cell type composition. By performing ScRNA-seq analysis of mESCs and gastruloids the authors identify the pathways that control the efficiency of gastruloid formation. The results are interesting and provide insights into the complexity of stem cell plasticity. However, in my opinion, there are important points that need to be addressed prior to publication.

We thank the reviewer for their very positive evaluation of our manuscript.

Major points:

The impact of different culture conditions on gastruloids formation have been previously reported. Unfortunately, the authors incorrectly cite the relevant literature. For instance, Cermola et al (Stem Cell Report 2021, ref.31) reported the effect of ESCs grown in different culture conditions//pluripotency state including but not limited to i) DMEM/FBS 15%/LIF, ii) N2B27 2i/LIF, and FGF/Activin -induced EpiLC and EpiSCs on gastruloids formation. They demonstrated that culture conditions significantly impact gastruloid formation, affecting i) cell aggregation (size of cell aggregates at 48 hours), ii) elongation (proportion of correctly elongated gastruloids), and iii) morphology (presence of aberrant gastruloids/multiple protrusions).

The authors should discuss their findings more thoroughly in the context of the current literature and properly cite the relevant papers. This will not diminish the significance of their studies.

We are happy that the reviewer appreciates the relevance of the presented work. We have adjusted and expanded our text to make sure existing literature is accurately cited and discussed. We have re-written the introduction to more carefully reference existing literature and included the reference of Anlas et al (Methods Mol Biol, 2021, PMID 33340359) where the impact of 2i pre-culture on gastruloid formation was reported on p10 line 239-240 and on p17 line 470. We have more accurately cited Cermola et al., and expanded the discussion on p17, lines 468-474 to include that different culture conditions between studies can also affect the reported gastruloid formation:

“In addition, another study using E14 mESCs reported increased efficiency of elongation after mESCs were maintained in 2i medium, and some cell lines require a 2i pulse for gastruloid formation (51, 53). It is therefore plausible that different cell lines require alternative pre-treatments to achieve the same receptive state as required for symmetry breaking during gastruloid formation. Additional experimental variations between studies, such as alterations in media composition and variations in the cell culture protocol, could also influence the reported differences.”

Materials and Methods

- Pag. 4, line 84: PBS0 and TrypLE are not defined. Please specify

- Pag. 5, lines 92-93: the concentration of 2i inhibitors (CHIR and PD) are expressed as %, but it is not clear % of what mass/vol?. Please clarify or indicate the molar concentration.

- Pag. 7, line 147: the sentence ‘Cells with either less than 500 reads and 100 genes detected in less than 2 cells were filtered out’. Please rephrase or clarify

This information was indeed missing in the methods description and is now added on page 5 line 108-109 (PBS0 and TrypLE) and p5 line 111-113 (CHIR and PD). Because two additional cell lines were included, the methods sections “cell culture and gastruloid generation” and “bright-field microscopy and analysis” (p4-6) were expanded. The sentence “Cells with either less than 500 reads and 100 genes detected in less than 2 cells were filtered out” was rephrased to “Cells with less than 500 reads or less than 100 genes detected were filtered out.” on p8, line 183.

Results

Pag. 8 lines 195-196: the authors claim that ..”pluripotency states in between naïve and ground state can be produced with short-term pulses of 2i and ESLIF”. However, this conclusion lacks experimental evidence. The authors should provide at least some characterization of the cells defined as “intermediate” states of pluripotency (conditions named 3, 4, 5 and 6) between conditions 1 and 2.

We agree that the microscopy-focused analysis in Fig1 is indeed not sufficient to prove the existence of intermediate states of pluripotency, rather than different mixtures of defined states. Therefore, we now expanded the single-cell differential gene expression and H3K27me3 analysis in Figs 2D-G, S3F and S4I. Together, these analyses show pre-culture specific transcription and chromatin states, also for intermediate states tested. Especially the single-cell data supports the statement of different cell states rather than different state mixtures.

Figure 1, pre-culture conditions, including the number of cells plated/density, the type of plates and coating used, are not specified.

We thank the reviewer for this comment. We now added the missing details about plates and cell densities used to the methods section on p4, line 99 (“gelatin-coated 6-well plate cell culture dishes”) and p5 line 104 (“Cells were split every second day at 80% density“). We also elaborated on the culture conditions used before gastruloid aggregation on p5, lines 118-119: “two days before aggregation, cells were plated in a series of 1:10 to 1:3 dilution, and at the time-point of aggregation the cells with 80% confluency were chosen”

Figure 1, panel A’. At the high cell density, assessing the effect of culture conditions on cell colony morphology seems difficult to evaluate/quantify. Therefore, the phenotype-based analysis of the intermediate states proposed here lacks convincing evidence. On the other hand, a molecular-based approach, i.e. the identification of specific markers for the 6 conditions, for instance based on data from the RNAseq analysis, seems not sufficient per se (see below).

We thank the reviewer for this comment. From what we understand from this remark, it seems this comment is based on a misperception due to the low magnification of the shown images. We now changed the images in Fig 1A’ for higher magnification examples that allow a clearer recognition of condition-specific colony morphologies. We also provide representative images of the newly included cell lines in Fig S1A. We have addressed the suggestion for the molecular-based approach in the points below, where we explain the expanded scRNA-seq analysis of the six pre-culture conditions in mESCs presented in Fig 2D-G and Fig S3D. We further elaborate on the differences in gene expression between all conditions.

Pag. 9 line 204. …”cells from all pre-culture conditions except ‘condition 2’ generate gastruloids efficiently”. The 2i medium is routinely used to capture naïve pluripotency, and so, the observation that ESCs grown in 2i medium (condition 2) are unable to elongate is unexpected. It is likely that 2i ESCs are ‘locked” in the naïve pluripotent state and that this eventually delay/prevent exit from pluripotency and symmetry breaking. Indeed, condition 5, that is 48h in LIF medium is sufficient to rescue the elongation efficiency (see also Discussion section below). Please, discuss.

Indeed, the B6 cell line shows a decreased axis ratio (AR) after 2i incubation, which contradicts previous observations mentioned by the reviewer. We have now added two more cell lines from a different genetic background, which show different dynamics. As a matter of fact, the IB10 line shows an increased AR after 2i culture. We therefore show that the effect of pre-culture conditions on gastruloid formation (and more specifically their AR) is cell line-dependent. We agree that it is possible that 2i ESCs from the B6 line are ‘locked in’ a naive pluripotent state, which could delay their exit from pluripotency and thereby differentiation. We have included an additional analysis in Figures 3F-H and S4G-H to address the question of developmental delay between pre-culture conditions, based on a suggestion by reviewer 2 (see discussion on heterochrony below). In this analysis, we did not observe evidence of developmental delay between the conditions tested, but it should be pointed out that we could only compare pre-culture conditions 1, 5, and 6. While we are not able to conclude whether condition 2 would result in a developmental delay in the B6 line, the results of this analysis underline the reviewer’s comment that in condition 5, a 48h pulse in ESLIF medium is sufficient to result in timely differentiation. We further hypothesise that it is possible that the ‘default’ pluripotency state differs per cell line and genetic background, and this in part explains their different responses to the pre-culture conditions tested as we now show in our manuscript. We have now included this in the discussion on page 17, lines 463-474:

“Here, we show that by modulating the mESC state through pre-culture media composition, the elongation length and reproducibility of gastruloid formation is affected. We first tested three different cell lines of various genetic backgrounds and noticed that the pre-culture condition resulting in the highest AR differs per cell line, highlighting the need for cell line-specific optimisation of the gastruloid protocol. While the B6 background yielded the lowest AR of gastruloids after 2i culture, this was not replicated in the TR and IB10 cell lines. In addition, another study using E14 mESCs reported increased efficiency of elongation after mESCs were maintained in 2i medium, and some cell lines require a 2i pulse for gastruloid formation. It is therefore plausible that different cell lines require alternative pre-treatments to achieve the same receptive state as required for symmetry breaking during gastruloid formation. Additional experimental variations between studies, such as alterations in media composition and variations in the cell culture protocol, could also influence the reported differences.”

Figure 1, the values of the elongation index are overall lower than those previously reported by different laboratories (2.5 or higher), and the abnormal value (see Fig.1, panel D, conditions 3, 4, 6 replicate B) appears to correspond to the diameter rather than the length. Indeed, the diameter is larger than expected (100-200 microM). Likely, it is due to the use of the largest circle (from Max Inscribed Circles Fiji plug-in) as the gastruloid diameter.

To avoid a potential miscalculation due to the usage of the largest circle, we improved the analysis using the MOrgAna software, as this estimates the length and width of gastruloids more accurately than the approach of using the diameter of the largest circle. Of note, this change in indexing of the gastruloids did not yield significant changes to the elongation index, confirming that the presented quantification does represent the imaging data. Having now included two additional cell lines we see larger elongation indexes for the IB10 cell line that are more similar to previous reports. We therefore conclude that the observed differences are cell line-specific.

Pag. 10 line 242-243, ScRNAseq analysis (Figure 2) revealed that ESCs even if grown in 6 different conditions, display only two different transcriptional and epigenetic signatures. These results raised some issues:

a.- What about the intermediates states of pluripotency. Is the ScRNAseq analysis able to discriminate the cells grown in the 6 conditions? The cells pre-grown in the 6 different conditions display significant differences in their transcriptome? Please clarify

We agree that the intermediate states of pluripotency presented in this manuscript were not highlighted sufficiently, and therefore included a more detailed analysis of the condition-specific differences in Fig 2D-G, including differentially expressed genes and a biological process enrichment analysis (p13, lines 342-349). We also expanded the analysis of H3K27me3 coverage in these conditions in figure S3F (p14-15, lines 382-385). Based on the differential gene expression analysis (Fig 2D-G) we now see that condition 1 is primed towards an ectoderm fate, but conditions 5 and 6 are not. Down-regulated genes in condition 2 compared to conditions 3 and 4 are enriched for GO terms of organismal development, indicating that conditions 3 and 4 display a less naive stage of pluripotency than condition 2. The expanded H3K27me3 analysis shows a loss of H3K27me3 on transcriptionally upregulated genes of the same condition.

b. Is the transcriptome profile relevant for the gastruloid formation efficiency?. For instance, cells from conditions 1, 5, and 6 seems to segregate together from the seq analysis but display a completely different gastruloid formation efficiency (Figure 1).

It seems that the presence of LIF for the last 24 hour (compare conditions 1, 5, and 6) is sufficient to induce a similar signature/cell identity (transcriptome and epigenetic signature).

We thank the reviewer for raising this interesting issue. As shown in figures 2D-G and S2F that we now expanded based on this comment, we can see ectoderm-specific genes getting expressed in condition 1 mESCs. This is consistent with the scRNAseq dataset of gastruloids (figure 3), where we see a higher ratio of ectodermal cells in condition 1 gastruloids. This indicates that the transcriptome of the mESC might be a predictive factor. Beyond this, our analysis could not identify further germ layer commitment by any of the conditions. This could potentially relate to the time points of the analysis, since we compare mESCs with 120 h AA gastruloids and this study does not include time points in between these states. It is possible that these subtle differences observed in mESCs are enlarged around the time point of WNT activation.

Pag. 13 line 323 to Pag. 14 line 346, ScRNAseq analysis (Figure 3) revealed that gastruloids generated with cells grown under different conditions (1, 5, and 6), have a different lineage compositions. These interesting results raise some issues:

a.- Is the composition of the gastruloids analyzed here (mainly condition in line with that reported in the literature with respect to types/number of cell lines identified, etc.?

The results text was adjusted to include a comparison of the observed differentiation biases with relevant literature. Cell types were annotated based on established marker genes shown in existing literature (page 16, lines 420-421). We elaborate on the comparison of cell type composition reported in Rosen et al., 2022, in the discussion on p18, lines 481-488:

“The gastruloid protocol was also reported to result in inter-gastruloid heterogeneity, with individual gastruloids being biased towards either an ectoderm or a mesoderm fate (22). This bias in specification was caused by differences in response strength to the Wnt-activation pulse. To take inter-gastruloid variability into account, we profiled 5 individual gastruloids per pre-culture and compared lineage contributions between replicates and between conditions. We did not observe similar inter-gastruloid heterogeneity in this dataset. Future studies are required to shed further light on how cellular heterogeneity in responsiveness to extracellular signals contributes to lineage decisions.”

b.- the high fraction of mesoderm in gastruloids derived from conditions 5 and 6 correlates with the induction of mesodermal markers in ESCs grown under these conditions?

In the new figure panels Fig 2D-E we indeed see the mesodermal gene Tbx3 as differentially expressed. However, more likely than c5/c6 being primed for mesoderm is that condition 1 is primed for ectoderm instead, indicated by marker gene expression in Fig S2. This is also indicated by condition 1-specific Shh signalling. The text was adjusted to address this question (p13 lines 342-349, p15-16 lines 410-414, p16 422-425).

Discussion

Pag. 14, lines 362-364. …”While in our study 2i grown mESCs yielded the lowest efficiency of gastruloid elonga

---

## [Decision Letter · Decision Letter 1]

2 Oct 2024

PONE-D-24-20935R1Stem cell culture conditions affect in vitro differentiation potential and mouse gastruloid formationPLOS ONE

Dear Dr. Blotenburg, Thank you for submitting your manuscript to PLOS ONE. After careful consideration, we feel that it has merit but does not fully meet PLOS ONE’s publication criteria as it currently stands. Therefore, we invite you to submit a revised version of the manuscript that addresses the points raised during the review process.

Both Reviewers and I are appreciative of the amount of time and effort put into this extensive revision. As you will see from the comments below, they are mostly positive, but there remain some concerns about the significance of the data presented, which I am keen for you to address to some extent. However, I am keen to keep revisions within the scope of the original reviews and requests for clarification or validation.

It is my view that the results presented are important for comparability of mouse gastruloid protocols and experimental results, and have been strengthened by the revisions. It is equally important to anchor the results on objective and transferable reference markers.

In that regard, I would like the Authors to address 2 remaining major points raised by Reviewer 2.

1. Discuss the implications for mouse gastruloid formation of long-term 2i culture conditions for ES cell maintenance.

2. Perform Brachyury staining at 48-72h to assist in the deconvoluting the impact of ES culture conditions and the timing of CHI pulsing in the differences observed at 120h.

Also, please make sure that you systematically include references for pathways, cell and tissue annotations.

We look forward to receiving your revised manuscript.

Kind regards,

Cristina Pina, MD DPhil

Academic Editor

PLOS ONE

Journal Requirements:

Reviewers' comments:

Reviewer's Responses to Questions

**Comments to the Author**

1. If the authors have adequately addressed your comments raised in a previous round of review and you feel that this manuscript is now acceptable for publication, you may indicate that here to bypass the “Comments to the Author” section, enter your conflict of interest statement in the “Confidential to Editor” section, and submit your "Accept" recommendation.

Reviewer #1: All comments have been addressed

Reviewer #2: (No Response)

2. Is the manuscript technically sound, and do the data support the conclusions?

Reviewer #1: Yes

Reviewer #2: Partly

3. Has the statistical analysis been performed appropriately and rigorously?

Reviewer #1: Yes

Reviewer #2: Yes

4. Have the authors made all data underlying the findings in their manuscript fully available?

Reviewer #1: Yes

Reviewer #2: Yes

5. Is the manuscript presented in an intelligible fashion and written in standard English?

Reviewer #1: Yes

Reviewer #2: Yes

6. Review Comments to the Author

Reviewer #1: (No Response)

Reviewer #2: I appreciate the authors' efforts in conducting some of the additional experiments and analyses I suggested, particularly the evaluation of the impact of distinct culture conditions on gastruloid morphology using different cell lines. These new results and the corresponding revisions have enhanced the manuscript, providing some clarity to the study. However, some issues are still unresolved and further work is required before the manuscript is ready for publication.

Main issues:

- The transcriptomic and epigenetic investigations remain limited to one cell line. Given the high variability found between cell lines in the initial culture conditions, it is important to extend these analyses to more cell lines and provide a thorough discussion of the results. In this discussion, the authors should also explain why longer periods of 2i treatment - recommended by laboratories such as that of Austin Smith to achieve a ‘tabula rasa’ state - were not considered for this study.

- The manuscript still lacks sufficient evidence showing whether the different culture conditions alter the cell state of the gastruloids at 48 hours (before CHIR treatment), and if this causes the observed differences at 120 hours. It is critical to provide this data to interpret the CHIR timing shift correctly. For instance, is symmetry breaking delayed when 2i is used? The rationale for altering the CHIR treatment timing needs to be clearly explained, with evidence. As previously suggested, stainings for key primitive streak markers like Brachyury between 48 and 72 hours would be a solution.

Minor issues:

- It should be visually clear which results in Figure 1 correspond to which cell line. The same applies to the new figures.

- The data suggesting the involvement of Shh signalling (“ectodermal priming signature with the expression of Gibx2, Ptch1, Sox11, and Lefty1, indicative of Sonic Hedgehog (Shh) signalling”) is insufficient and lacks appropriate references. Additionally, the role of this signalling in pluripotency should be discussed.

- The authors’ annotation of the different cell types and pluripotency states has improved but still needs refinement. For example, is Tbx4 a mesodermal marker? I suggest the authors review each marker mentioned in the text and provide proper references.

- The issue of potential heterochrony should be resolved by including a heatmap showing the expression of Hox genes across the different conditions.

- The authors should lower their claims regarding the novelty of their screening approach as similar methodologies have been published already by other laboratories (e.g. Lutolf’s lab).

7. PLOS authors have the option to publish the peer review history of their article (what does this mean? ). If published, this will include your full peer review and any attached files.

**Do you want your identity to be public for this peer review?** For information about this choice, including consent withdrawal, please see our Privacy Policy .

Reviewer #1: No

Reviewer #2: No

---

## [Author Response · Author response to Decision Letter 2]

21 Dec 2024

Point-by-point response to the reviewer comments:

Reviewer #2: I appreciate the authors' efforts in conducting some of the additional experiments and analyses I suggested, particularly the evaluation of the impact of distinct culture conditions on gastruloid morphology using different cell lines. These new results and the corresponding revisions have enhanced the manuscript, providing some clarity to the study. However, some issues are still unresolved and further work is required before the manuscript is ready for publication.

We thank the reviewer for their positive evaluation of the revised manuscript.

Main issues:

- The transcriptomic and epigenetic investigations remain limited to one cell line. Given the high variability found between cell lines in the initial culture conditions, it is important to extend these analyses to more cell lines and provide a thorough discussion of the results.

Indeed, the results of these proposed experiments would be highly interesting. Thanks to the previous reviewer comments and resulting manuscript revisions, we have included three cell lines in the first part of this study. This has generated important understanding on the effect of short-term pre-culture conditions on differentiation in different genetic backgrounds. While we would be curious to explore the transcriptome and epigenetic states of these different cell lines as described in the second part of this study for the B6 line, this would extend the contents of the manuscript far beyond our currently available time frame and resources.

In this discussion, the authors should also explain why longer periods of 2i treatment - recommended by laboratories such as that of Austin Smith to achieve a ‘tabula rasa’ state - were not considered for this study.

The work of Austin Smith and others regarding 2i treatment on mESCs has been highly influential and this state of long-term 2i culture has been extensively studied. Importantly, there are considerable differences between short- and long-term 2i culture in regards to the epigenetic and transcriptomic state of the cells. Upon growing gastruloids, we noticed that even a 24-hour pulse of 2i already considerably changed the morphology of the mESC starting culture and resulting gastruloids. For the purpose of this manuscript we therefore decided to focus on the effect of short-term pulses, with the hypothesis that subtle changes in pluripotency state affect downstream differentiation programs and would aid in defining a protocol with increased stability and reproducibility. To our knowledge, the use of short-term pulses of 2i has been less studied than the clearly defined long-term 2i state. We have added this statement to the discussion, p19, lines 512-514:

Since the impact of such long term 2i treatment on ES cell differentiation state and gastruloid formation was already thoroughly studied, in this work we focused on shorter 2i pulses to adjust and synchronise their cellular states.

The manuscript still lacks sufficient evidence showing whether the different culture conditions alter the cell state of the gastruloids at 48 hours (before CHIR treatment), and if this causes the observed differences at 120 hours. It is critical to provide this data to interpret the CHIR timing shift correctly. For instance, is symmetry breaking delayed when 2i is used? The rationale for altering the CHIR treatment timing needs to be clearly explained, with evidence.

After careful re-evaluation of the paragraph on p.12, lines 298-304, we think we may have inadvertently caused some confusion. The main and intended message of this paragraph is that in Supp Fig G-I we found no significant evidence that a shift in timing of the Chiron pulse improved gastruloid elongation, with the exception of condition 3. While this was important to control, all other experiments in the manuscript therefore were performed with the standard Chiron pulse (48-72 hours). We apologise that we did not formulate our description and conclusions from these experiments more clearly and have made the following textual changes to reflect this:

p.12 lines 298-304 previously read: “Because the timing of the chiron pulse could influence gastruloid formation, we subjected all conditions additionally to a delayed chiron pulse from 72 h - 96 h AA, and assessed them at 120h or 144 h AA (Fig S1G-I). While gastruloids from most pre-culture conditions show a lower AR following the delayed chiron pulse, condition 3 cells display a significant increase in AR after a delayed pulse and delayed read-out (Figs S1H-I). Therefore, we subjected condition 3 pre-culture cells to a 72 h - 96 h AA chiron pulse and read-out at 144 h AA, while for all other pre-culture mESCs we used the default gastruloid protocol.”

In the current, revised version, we now included: “Because the altered pluripotency state of the different pre-culture conditions could have an effect on symmetry breaking during the gastruloid protocol, we subjected all conditions additionally to a delayed chiron pulse from 72 h - 96 h AA, and assessed them at 120h or 144 h AA (Fig S1G-I). We only observed a significant increase of the aspect ratio for pre-culture condition 3 with a 24h delay of the chiron pulse and read-out (Figs S1H-I). Therefore, we subjected condition 3 pre-culture cells to a 72 h - 96 h AA chiron pulse and read-out at 144 h AA, while for all other pre-culture mESCs we used the default gastruloid protocol.”

As previously suggested, stainings for key primitive streak markers like Brachyury between 48 and 72 hours would be a solution.

We thank the reviewer for this comment, which indeed helps further interpretation of our findings. We have added the proposed experimental data in Fig S1J-K and describe the results on p12, lines 304-309:

To investigate what causes this difference between pre-cultures, we assessed the time point of Brachyury (T) induction and polarisation between pre-cultures (Fig S1J-K). We found that while condition 1 results in polarised structures at 72h AA, conditions 2 and 3 display a delay in T induction and polarisation. Condition 4 pre-cultured gastruloids display similar symmetry breaking as compared to condition 1, but with a lower overall expression of T. At 96h AA, all pre-culture conditions result in elongated structures with polarised expression of T.

Minor issues:

- It should be visually clear which results in Figure 1 correspond to which cell line. The same applies to the new figures.

We have clarified the cell lines used within figure panels 1A’, 1D-E, and S1D-I. Figures 2, 3, S2-4 were generated with the B6/129 mESC cell line only, and we have clarified this in the titles of the figure descriptions. This is also specified in the text, p.12 lines 295-297: We decided to study the effect of all pre-culture conditions in more detail in the B6 cell line to see if other conditions could lead to more consistent results across replicates.

- The data suggesting the involvement of Shh signalling (“ectodermal priming signature with the expression of Gibx2, Ptch1, Sox11, and Lefty1, indicative of Sonic Hedgehog (Shh) signalling”) is insufficient and lacks appropriate references. Additionally, the role of this signalling in pluripotency should be discussed.

We thank the reviewer for this comment, and apologize for the lack of appropriate references for this statement. This statement is now rewritten for clarity (p.14, lines 361-363: “...and condition 1 mESCs show expression of factors associated with Sonic Hedgehog (Shh) signalling, such as Gbx2, Ptch1, Sox11, and Lefty1, indicative of ectodermal priming (Fig S2F)”). Appropriate references (1) identifying Gbx2, Ptch1, Sox11, and Lefty1 as Shh pathway components and (2) implicating Shh signalling in ectodermal specification are now provided.

- The authors’ annotation of the different cell types and pluripotency states has improved but still needs refinement. For example, is Tbx4 a mesodermal marker? I suggest the authors review each marker mentioned in the text and provide proper references.

We have provided references for each (list of) used marker genes. We apologise for the confusion surrounding Tbx4, indeed it has been used as a marker gene for allantoic mesoderm (see references 15, 53) and we have now specified this accordingly in the manuscript.

- The issue of potential heterochrony should be resolved by including a heatmap showing the expression of Hox genes across the different conditions.

We have included the requested figure panel in S4J.

- The authors should lower their claims regarding the novelty of their screening approach as similar methodologies have been published already by other laboratories (e.g. Lutolf’s lab).

We now highlight previous work using screening approaches to improve in vitro differentiation protocols in the discussion (p20, lines 537-539):

Importantly, building upon previous work to systematically improve gastruloid efficiency and complexity (71,72), we present an unbiased screening approach to assess the efficiency of gastruloid formation.

---

## [Editor Report · Decision Letter 2]

26 Dec 2024

Stem cell culture conditions affect in vitro differentiation potential and mouse gastruloid formation

PONE-D-24-20935R2

Dear Dr. Blotenburg,

We’re pleased to inform you that your manuscript has been judged scientifically suitable for publication and will be formally accepted for publication once it meets all outstanding technical requirements.

Kind regards,

Cristina Pina, MD DPhil

Academic Editor

PLOS ONE

---

## [Editor Report · Acceptance letter]

PONE-D-24-20935R2

PLOS ONE

Dear Dr. Blotenburg,

I'm pleased to inform you that your manuscript has been deemed suitable for publication in PLOS ONE. Congratulations! Your manuscript is now being handed over to our production team.

Kind regards,

on behalf of

Dr. Cristina Pina

Academic Editor

PLOS ONE